Corrected: Publisher correction

# ID3 regulates the MDC1-mediated DNA damage response in order to maintain genome stability

Jung-Hee Lee[1,2], Seon-Joo Park[1,3], Gurusamy Hariharasudhan[1], Min-Ji Kim[1], Sung Mi Jung[1], Seo-Yeon Jeong[1,2], In-Youb Chang[4], Cheolhee Kim[5], Eunae Kim [5], Jihyeon Yu[6], Sangsu Bae[6] & Ho Jin You[1,7]

MDC1 plays a critical role in the DNA damage response (DDR) by interacting directly with several factors including γ-H2AX. However, the mechanism by which MDC1 is recruited to damaged sites remains elusive. Here, we show that MDC1 interacts with a helix–loop–helix (HLH)-containing protein called inhibitor of DNA-binding 3 (ID3). In response to double-strand breaks (DSBs) in the genome, ATM phosphorylates ID3 at serine 65 within the HLH motif, and this modification allows a direct interaction with MDC1. Moreover, depletion of ID3 results in impaired formation of ionizing radiation (IR)-induced MDC1 foci, suppression of γ-H2AX-bound MDC1, impaired DSB repair, cellular hypersensitivity to IR, and genomic instability. Disruption of the MDC1–ID3 interaction prevents accumulation of MDC1 at sites of DSBs and suppresses DSB repair. Thus, our study uncovers an ID3-dependent mechanism of recruitment of MDC1 to DNA damage sites and suggests that the ID3–MDC1 interaction is crucial for DDR.

[1] Laboratory of Genomic Instability and Cancer Therapeutics, Cancer Mutation Research Center, Chosun University School of medicine, Gwangju 501-759, Republic of Korea. [2] Department of Cellular and Molecular Medicine, Chosun University School of medicine, Gwangju 501-759, Republic of Korea. [3] Department of Premedical Sciences, Chosun University School of medicine, Gwangju 501-759, Republic of Korea. [4] Department of Anatomy, Chosun University School of medicine, Gwangju 501-759, Republic of Korea. [5] College of Pharmacy, Chosun University, 375 Seosuk-dong, Gwangju 501-759, Republic of Korea. [6] Department of Chemistry, Hanyang University, Seoul 04763, Republic of Korea. [7] Department of Pharmacology, Chosun University School of medicine, Gwangju 501-759, Republic of Korea. Jung-Hee Lee and Seon-Joo Park contributed equally to this work. Correspondence and requests for materials should be addressed to J.-H.L. (email: jhlee75@chosun.ac.kr) or to H.J.Y. (email: hjyou@chosun.ac.kr)

The integrity of genomic DNA is challenged by genotoxic insults that originate from either normal cellular metabolism or external sources. To ensure proper maintenance of genomic integrity, eukaryotes have evolved a DNA damage response (DDR) system that senses damage and transduces this information within the cell in order to orchestrate DNA repair, cell-cycle checkpoints, chromatin remodeling and apoptosis[1]. The functional importance of DDR in maintaining genomic integrity is highlighted by the fact that it is conserved among eukaryotes. Mutations that disrupt the activity of DDR components contribute directly to tumorigenesis[2]; therefore, it is important to understand these complex mechanisms at the molecular level to further our understanding of cancer progression and treatment.

DNA double-strand breaks (DSBs), which are generated through ionizing radiation (IR) and through various DNA-damaging chemicals, are the most dangerous DNA lesions, because if they are not efficiently and accurately repaired, they can result in mutations, genomic rearrangements, and cell death, which can lead to cancer[1, 2]. The ability of cells to detect and properly repair DSBs is therefore essential for maintaining genome stability and preventing cancer[3]. Central to the DSB checkpoint response is ATM protein kinase, which, when activated by DSBs, initiates a signaling cascade that starts with phosphorylation of the histone variant H2AX (γ-H2AX) at DSB sites, and is followed by recruitment of upstream factors including MDC1[1, 4, 5]. MDC1 functions as an assembly platform to help localize and maintain signaling and repair factors at and around DSB sites[6]. In this role, MDC1 amplifies DNA damage signals by binding to phosphorylated H2AX and subsequently binding and retaining additional DDR factors at sites of DNA damage. The accumulation of these DDR factors at DSB sites is generally believed to facilitate DNA damage repair and checkpoint control. Thus, MDC1 has been recognized as the "master regulator" that modulates a specific chromatin microenvironment required to maintain genomic stability.

MDC1-knockout (KO) mice show chromosomal instability, defects in DSB repair, radiosensitivity, and cancer predisposition[7, 8]. Furthermore, downregulation of MDC1 is associated with multiple cellular phenotypes including hypersensitivity of cells to DSBs, improper activation of the G2/M and intra-S checkpoints, aberrant activation of DNA damage-induced apoptosis, and inefficient phosphorylation of DDR regulatory proteins[9]. It has been suggested that, in addition to its central role in the DDR, MDC1 directly mediates HR[10, 11] and non-homologous end joining (NHEJ)[12], activation of the decatenation checkpoint[13], regulation of the DNA replication checkpoint[14], mitosis[15], and spindle assembly checkpoint[16].

Clearly, MDC1 is quickly recruited to DNA damage sites, allowing multiple protein–protein interactions that are crucial for proper DDR processes. However, the precise mechanisms by which MDC1 is recruited to protect cells from the deleterious effects of DNA damage are not fully understood. The current study was initiated with the goal of better understanding how MDC1 is recruited to DNA damages sites and how the role of MDC1 in DDR is regulated in response to DNA damage. Since a tandem BRCA1 C-terminal (tBRCT) domain of MDC1 is essential for recruitment of MDC1 to DNA damage sites[17], we screen for tBRCT domain of MDC1-associated proteins and identify a helix–loop–helix (HLH) domain-containing protein called inhibitor of DNA-binding 3 (ID3), which we propose interacts directly with MDC1 and is a key factor in the interaction of MDC1 with γ-H2AX, recruiting MDC1 to DSB sites and regulating DDR function of MDC1.

## Results

**MDC1 interacts with ID3.** Although the role of mammalian MDC1 in the DDR is well documented, its regulation and underlying mechanism of action are only partially understood. In order to better characterize the regulatory network relevant to MDC1 and to gain further insight into the molecular mechanism of action of MDC1 in the DDR, a yeast two-hybrid screen was performed using a HeLa cDNA plasmid library with the C-terminal fragment (amino acid 1882–2082) of human MDC1 as the bait. Out of the $2.6 \times 10^7$ transformants that were screened, 43 independent positive clones were isolated. When cDNA from each of the positive clones was sequenced, one was identified as 53BP1, a protein known to associate with the tandem BRCA1-C terminus (tBRCT) domain of MDC1. The other clones, which had no previously identified association with MDC1, encoded ID3 (NM_002167), PIAS1 (NM_016166), UBE2I (NM_194259), KPNA2 (NM_0022266), ZNF114 (NM_153608), KIFC1 (NM_002263), CASP8AP2 (NM_001137667), C2orf44 (NM_025203), SRSF11 (NM_001190987), TINP1 (NM_014886), GPRC5C (NM_022036), and WWC1 (NM_015238). Among these, DNA-binding protein inhibitor ID3 was particularly notable because this HLH-containing protein has been shown to activate a DNA repair process[18] and, correspondingly, when ID3 is inactivated, excess DNA damage accumulates[19].

To verify that an interaction between MDC1 and ID3 occurs in human cells, we used co-immunoprecipitation assays followed by western blotting to assess protein–protein interactions. As shown in Fig. 1a, b, endogenous MDC1 and ID3 co-immunoprecipitated reciprocally and, although the association occurred in non-irradiated cells, it was enhanced in response to DNA damage. To examine this further, we expressed HA-tagged MDC1 and GFP-tagged ID3 ectopically in HEK293T cells and attempted to immunoprecipitate MDC1 using anti-HA antibody. As shown in Fig. 1c, GFP-ID3 was observed in the anti-HA immunoprecipitate. Reciprocally, HA-MDC1 was immunoprecipitated together with GFP-ID3 using anti-GFP antibody (Fig. 1d) and these interactions were more pronounced following exposure to IR. This protein–protein interaction did not require the presence of DNA, as neither ethidium bromide nor DNase affected the co-immunoprecipitation (Supplementary Fig. 1a). Furthermore, ID3 appears to be the only member of the ID family of proteins with which MDC1 interacts, because we were unable to detect any interaction of MDC1 with ID1 or ID2 (Supplementary Fig. 1b).

To confirm that the tBRCT domain of MDC1 is the location at which ID3 binds, we examined the interaction between these two proteins using a series of internal deletion mutations of MDC1 (Supplementary Fig. 2a). We found that when expressed in HEK293T cells, all MDC1 mutants co-immunoprecipitated with GFP-ID3 except for a mutant lacking tBRCT domain (amino acids 1893–2082) (Supplementary Fig. 2b), indicating that this domain is indeed required for binding to ID3. To further analyze direct physical associations between ID3 and MDC1, a GST fusion to the tBRCT domain of MDC1 was made and used in a pull-down assay with IR-treated HeLa cell lysates. Immunoblotting of the pull-down samples revealed that the tBRCT domain of MDC1 interacts with ID3 (Fig. 1e); this polypeptide also pulled down the exogenously expressed GFP-tagged ID3 (Fig. 1f), confirming that ID3 directly binds to this region of MDC1 in vitro.

**ATM-dependent and MDC1-dependent phosphorylation of ID3.** To examine the potential role of ID3 in regulating DDR, we asked whether IR exposure induced phosphorylation of ID3. ATM/ATR-like kinases convey signals upstream of the DDR pathway by preferentially phosphorylating substrate proteins on either a serine or threonine residue that is typically followed by a glutamine residue in the so-called serine–glutamine (SQ) or threonine–glutamine (TQ) motifs[20, 21]. We identified two

putative ATM/ATR phosphorylation motifs in the ID3 amino acid sequence, threonine 62 (Thr62) and serine 65 (Ser65), both located in the conserved HLH domain (Fig. 2a). A sequence alignment of the human ID family of proteins, including ID1, ID2, and ID3, revealed that ID3 is the only member that possesses TQ (including Thr62) and SQ (including Ser65) motifs in the Helix2 domain, consistent with our earlier observation that ID3 is the only of the three family members relevant to this pathway. To test whether DNA damage induces phosphorylation of ID3, we immunoprecipitated ID3 from HeLa cells and examined phosphorylation levels before and after cells had been exposed to IR. Using antibodies that bind to phosphorylated serine or threonine, we showed that in IR-induced cells, serine was phosphorylated, but threonine was not (Fig. 2b). Western blot analysis, using lysates prepared from both control and irradiated cells and polyclonal rabbit anit-pSer65–ID3 antibody, confirmed that IR treatment enhanced the phosphorylation of endogenous ID3 at Ser65 (Fig. 2c). Recognition of pSer65–ID3 by this antibody was blocked by the presence of phosphorylated peptides but not by non-phosphorylated peptides (Supplementary Fig. 3a) and the recognition was inhibited by ID3 siRNA (Supplementary Fig. 3b), findings that establish the specificity of this antibody.

Next, we explored upstream signaling molecules in order to identify those that may contribute to DNA damage-induced phosphorylation of ID3. ATM kinase and DNA-dependent protein kinase (DNA-PK) are largely responsible for initiating and maintaining DNA damage signals after exposure to IR[20]. We therefore used KU55933, an inhibitor of ATM kinase, and NU7026, an ID-PK to assess the importance of these two proteins

in this regulatory pathway. In IR-treated HeLa cells, phosphorylation at Ser65 was strongly inhibited by KU55933 but was not affected by NU7026 (Fig. 2d). Because MDC1 associates with ID3, we looked for a role for MDC1 specifically in the phosphorylation of ID3. We compared normal and MDC1-depleted HeLa cells and observed that IR-induced ID3 phosphorylation was attenuated when MDC1 was expressed at lower levels (Fig. 2e). Together, these results suggest that ATM and MDC1 act together in mediating phosphorylation of ID3 Ser65 in response to DNA damage in vivo.

**Recruitment of phosphorylated ID3 to sites of DNA damage.** The finding that ID3 is phosphorylated at residue Ser65 after exposure to IR suggests that this form of ID3 may localize to site of DNA damage. To identify the cellular location of phosphorylated ID3, we used immunofluorescence staining with anti-pSer65–ID3 antibody. As shown in Fig. 3a, in IR-treated HeLa cells, foci of pSer65–ID3 were detected, and these foci co-localized with γ-H2AX, an intracellular protein that associates with damaged DNA. The staining of pSer65–ID3 was specific, because the number of foci decreased in the presence of a competing phosphorylated peptide but not in the presence of a non-phosphorylated peptide (Supplementary Fig. 3c). Because both ATM and MDC1 affect the phosphorylation of ID3, we looked for a role for either of these proteins on pSer65–ID3 foci formation. In the presence of the ATM inhibitor KU55933, a few IR-induced pSer65–ID3 foci were observed, whereas in the presence of NU7026, the DNA-PK inhibitor that had no effect in a

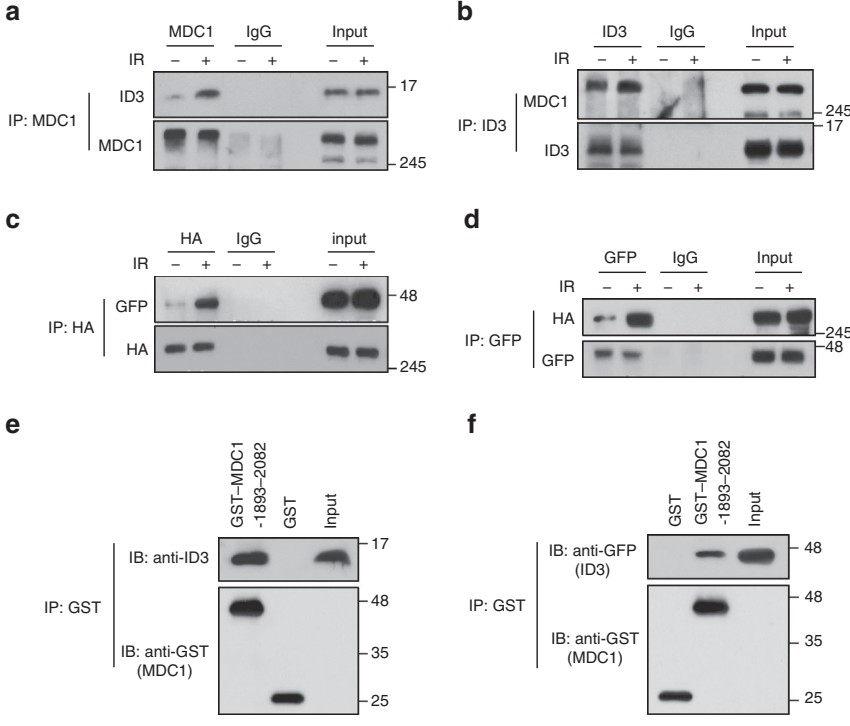

**Fig. 1** Protein–protein interactions between MDC1 and ID3. **a** HeLa cells, with or without exposure to IR, were collected after 3 h, and whole-cell lysates were subjected to immunoprecipitation using an anti-MDC1 antibody followed by western blotting using the antibodies indicated to the right of the blot. **b** HeLa cells were prepared as in **a**, and lysates were subjected to immunoprecipitation using an anti-ID3 antibody followed by western blotting using the antibodies indicated to the right of the blot. **c** HA-tagged MDC1 and GFP-tagged ID3 were co-transfected into HEK293T cells and exposed to IR. After 3 h, whole-cell lysates were subjected to immunoprecipitation using an anti-HA antibody followed by western blotting using the antibodies indicated to the right of the blot. **d** HEK293T cells were prepared as in **c**, and lysates were subjected to immunoprecipitation using an anti-GFP antibody followed by western blotting using the antibodies indicated to the right of the blot. **e**, **f** A GST-tagged fragment of MDC1 or a GST bead alone was incubated with total cell lysates from HeLa cells exposed to IR (**e**) or from GFP-ID3 transfected HEK293T cells exposed to IR (**f**). GST pull-downs were immunoblotted with antibodies as indicated. Uncropped blots of this figure accompanied by the location of molecular weight markers are shown in Supplementary Fig. 14

previous experiment, the number of foci was comparable to the control, as expected (Fig. 3b). Likewise, in MDC1-depleted cells exposed to IR, accumulation of pSer65–ID3 at DSBs was strongly reduced (Fig. 3c, d), supporting the prediction that both MDC1 and ATM are required for efficient localization of ID3 under these conditions.

**ID3 is required for the recruitment of MDC1 to DSBs**. The above results, pointing to a biochemical interaction between ID3 and MDC1, prompted the prediction that ID3 is required for recruitment of MDC1 to DSBs. To test this hypothesis, we exposed control and ID3 knockdown HeLa and U2OS cells (Fig. 4a) to IR, and used immunofluorescence staining to detect MDC1 at various time points. As shown in Fig. 4b, c, in control siRNA-transfected cells, MDC1 foci formed rapidly in response to IR treatment and the percentage of cells containing a notable number of foci increased steadily for up to 1 h. In contrast, cells transfected with ID3-targeted siRNA had significantly fewer MDC1 foci and the percentage of cells with foci remained low throughout the entire time course. We also observed that HeLa cells with a stable knockdown of ID3, created using two different shRNAs, had dramatically less recruitment of MDC1 to DNA damage sites (Supplementary Fig. 4a, b). Similar results were obtained when DNA damage was induced in HeLa cells using the radiomimetic neocarzinostatin (Supplementary Fig. 4c). Introduction of shRNA-resistant ID3 into cells depleted of endogenous ID3 restored formation of MDC1 foci, confirming that the

observed effects are attributed to the ID3 protein (Fig. 4d, e). MDC1 is required for the retention of additional DDR proteins and is thus considered to be an upstream regulator of this process[8,22,23]. Therefore, we predicted that depletion of ID3 might also affect the localization of several DDR factors to nuclear DSB sites. Indeed, accumulation of NBS1, BRCA1, 53BP1, RNF8, and RNF168 at DSBs in response to IR exposure was consistently impaired in both HeLa and U2OS cells depleted of ID3, whereas γ-H2AX, a protein that acts upstream in the DDR pathway, still formed foci (Fig. 4f, g).

To further confirm the critical role for ID3 in recruitment of MDC1 to nuclear foci after DNA damage, we generated *ID3*-KO human U2OS cells using the CRISPR/Cas9 genome-editing system (Supplementary Fig. 5a–c). Noticeably, IR-induced MDC1 foci were significantly reduced in *ID3* KO cells than in control cells (Supplementary Fig. 6a). Moreover, *ID3* KO impaired the recruitment of NBS1, BRCA1, 53BP1, RNF8, and RNF168, but not γ-H2AX, at DSBs after IR exposure (Supplementary Fig. 6b, c). Together, these data suggest that ID3 is required for localization of MDC1 at DSBs thereby facilitating recruitment of downstream factors.

**pSer65–ID3 is essential for IR-induced MDC1 foci formation**. Phosphorylation of SQ or TQ motifs leads to altered interactions between proteins in the DNA repair complex and between other checkpoint proteins[24]. Furthermore, proteins with BRCT domains, like MDC1, have been shown to recognize phospho-

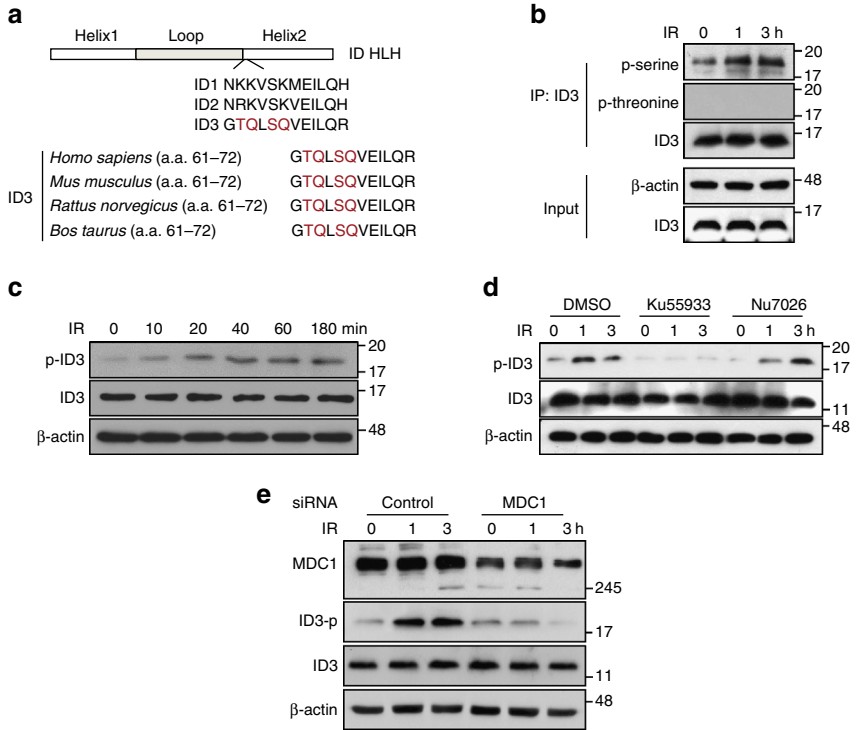

**Fig. 2** ATM- and MDC1-mediated phosphorylation of ID3 at Ser65 in response to IR. **a** An outline of the HLH domain and amino acid sequences including the TQ/SQ sites for human ID1, ID2, and ID3 is shown at the top of the panel. The bottom of the panel includes an alignment of ID3 HLH from four different species, demonstrating the highly conserved TQ and SQ motifs (highlighted). **b** HeLa cells, with or without exposure to IR, were collected at the indicated times and whole-cell lysates were subjected to immunoprecipitation using an anti-ID3 antibody followed by western blotting using anti-p-serine, anti-p-threonine, and anti-ID3 antibodies, as indicated to the right of the blot. **c** HeLa cells, with or without exposure to 2 Gy of IR for the indicated times, were lysed and analyzed by western blotting using anti-pSer65–ID3 and anti-ID3 antibodies. **d** HeLa cells were pretreated with DMSO, ATM inhibitor KU55933 (10 μM), or DNA-PK inhibitor NU7026 (5 μM) for 1 h, and then treated with or without exposure to IR for the indicated times. Cell lysates were analyzed by western blotting using anti-pSer65–ID3 and anti-ID3 antibodies. **e** HeLa cells transfected with either control siRNA or MDC1-specific siRNA, were treated with or without exposure to IR for the indicated times. Cell lysates were analyzed by western blotting using anti-MDC1, anti-pSer65–ID3, and anti-ID3 antibodies. Uncropped blots of this Figure accompanied by the location of molecular weight markers are shown in Supplementary Fig. 14

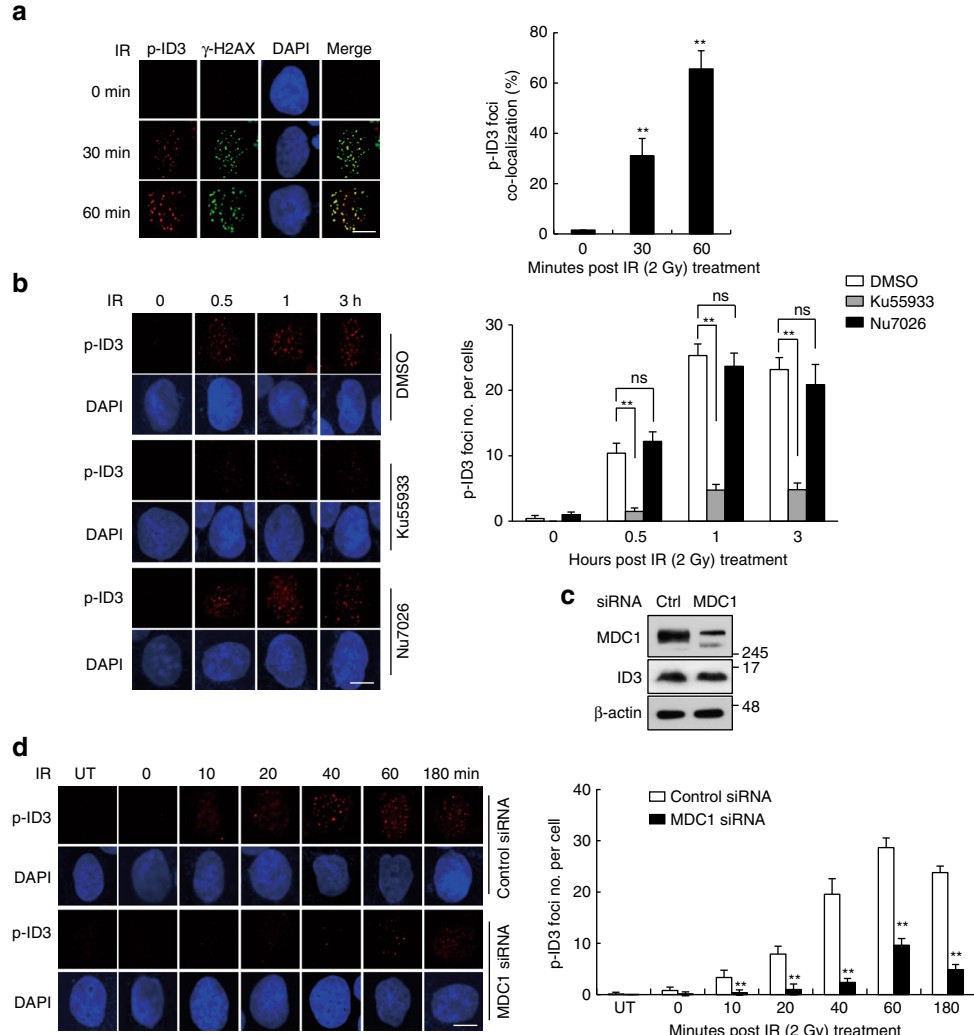

**Fig. 3** Phosphorylated ID3 is recruited to sites of DNA damage. **a** Endogenous phospho-ID3 co-localizes with γ-H2AX after DNA damage. HeLa cells were exposed to 2 Gy of IR and fixed at the indicated time points. Immunohistochemistry was performed using antibodies against pSer65–ID3 and γ-H2AX. Co-localization is visible as yellow staining in the column labeled "merge". Nuclei were stained with DAPI. The histogram in the right panel shows the percentage of pSer65–ID3 foci that co-localized with γ-H2AX foci. At least 100 cells were analyzed for each treatment (n = 3). Scale bars: 10 μm. **b** ATM, but not DNA-PK, mediates phospho-ID3 foci formation. HeLa cells were pretreated with DMSO, KU55933 (10 μM), or NU7026 (5 μM), exposed to 2 Gy of IR, and fixed at the indicated time points. Immunostaining experiments were performed using an anti-pSer65–ID3 antibody. The histogram in the right panel shows the number of phospho-ID3 foci per cells (n = 3). Scale bars: 10 μm. **c** HeLa cells were transiently transfected with either control siRNA or MDC1-specific siRNA, and the levels of endogenous MDC1 and ID3 were analyzed by western blotting. **d** HeLa cells transfected with either control siRNA or MDC1-specific siRNA were exposed to 2 Gy of IR and fixed at the indicated time points. Immunofluorescence was performed using antibodies against pSer65–ID3. The histogram in the right panel shows the number of phospho-ID3 foci per cell (n = 3). Scale bars: 10 μm. Data are presented as means ± s.d. P values are based on Student's two-tailed t-test: \*\*P < 0.01; ns, not significant

Ser/Thr motifs[25–27]. Therefore, we predicted that phosphorylation of Ser65 within the HLH domain of ID3 is important for modulating the interactions between MDC1 and ID3. To test this, we generated GFP-tagged constructs that included the following versions of ID3: the HLH domain (amino acids 25–81), a mutant lacking the HLH (ΔHLH; deletion of residues 25–81), a mutant lacking the carboxyl terminus (ΔC; deletion of residues 82–119), and a mutant lacking the amino terminus (ΔN; deletion of residues 1–24) (Fig. 5a). ID3 from each of these constructs was co-expressed with full-length HA-tagged MDC1 in HEK293T cells. As shown in Fig. 5b, when the HLH domain was present, ID3 interacted with MDC1, but if the HLH domain was deleted, the interaction was impaired.

We then generated a point mutation within the HLH domain of ID3, replacing Ser65 with alanine (S65A). As a negative

control, we exchanged Thr62 to alanine (T62A). As shown in Fig. 5c, the presence of the S65A mutation impaired interactions with MDC1, but the T62A mutation did not have a notable affect, suggesting that phosphorylation of ID3 at Ser65 is required for its interaction with MDC1. We next mutated Ser65 to aspartic acid (S65E), which mimics the phosphorylated state. This mutant bound to MDC1, as predicted (Fig. 5d). Interactions between ID3 and the tBRCT domain of MDC1 were abolished after treatment with phosphatase, supporting the conclusion that the interaction is phosphorylation dependent (Supplementary Fig. 7).

The BRCT domain of MDC1 is required for the formation of foci at sites of DNA damage[17]. Because ID3 interacts with this same domain, and because depletion of ID3 decreased the formation of MDC1 foci, we hypothesized that a direct interaction between ID3 and MDC1 is required for MDC1 to

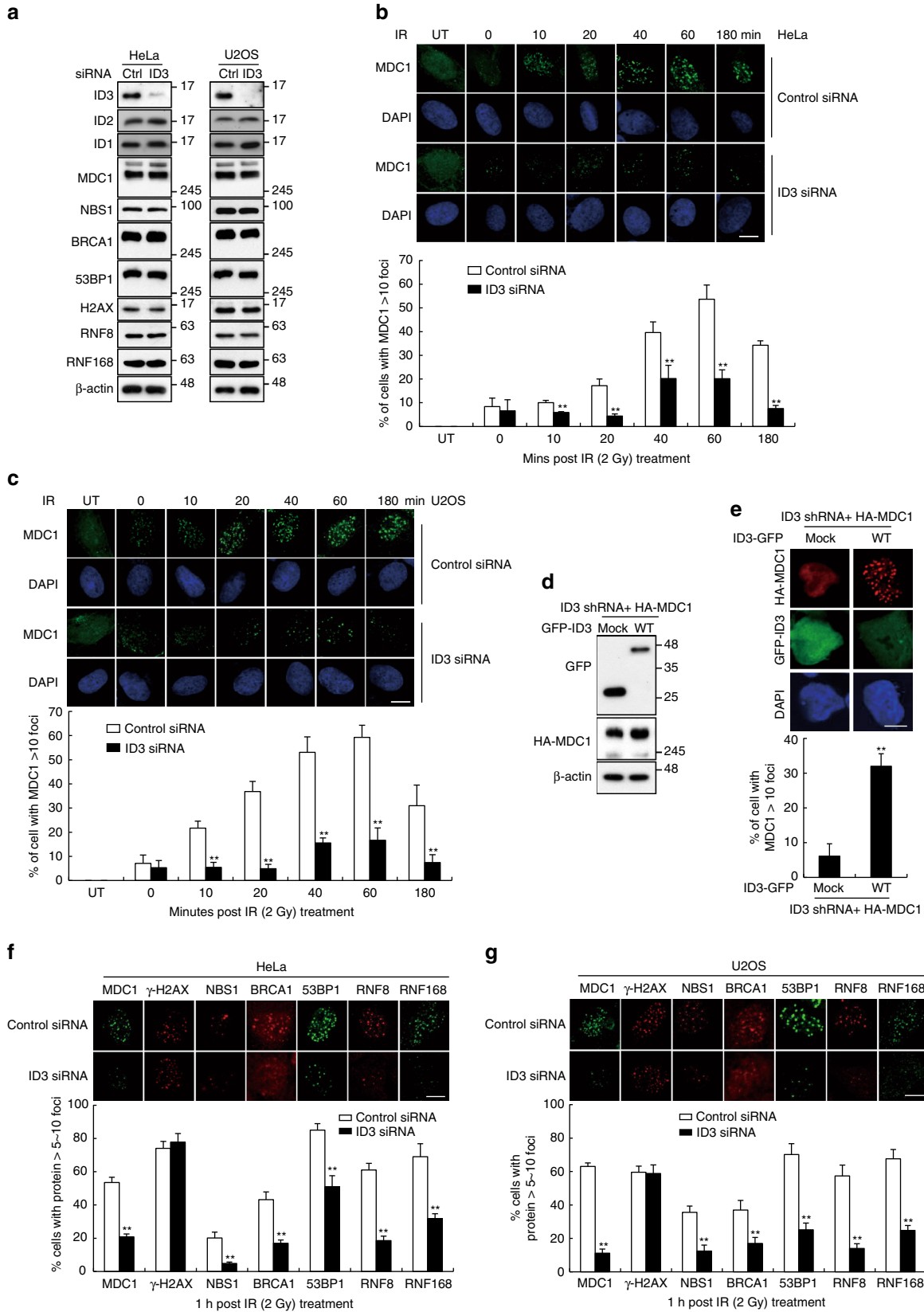

be recruited to sites of DNA damage. We tested this using HeLa cells that had been transfected with ID3 shRNA in order to deplete the endogenous pool of protein, and then reconstituted with either an shRNA-resistant HLH domain (HLH ID3) or an HLH deletion mutant (ID3-ΔHLH) (Fig. 5e). The level of

recruitment of HA-MDC1 to DSB sites in each cell type was then measured. As shown in Fig. 5f, there was significantly more recruitment of HA–MDC1 to DSBs in cells expressing HLH ID3 when compared to cells expressing ID3-ΔHLH, indicating that the HLH domain of ID3 is indeed involved in recruitment of

MDC1. Moreover, immunofluorescence analysis suggested that when ID3 knockdown cells are reconstituted with either wild-type HLH domain of ID3 or the T62A HLH mutant, recruitment of MDC1 to DSBs occurred normally. But when reconstituted with the S65A HLH mutant, recruitment was significantly decreased (Fig. 5g, h). On the other hand, reconstitution of ID3 knockdown cells with the S65E HLH mutant, which mimics the phosphorylated state, restored MDC1 recruitment to normal levels. These results point to a requirement for a phospho-dependent interaction between MDC1 and ID3 in order for proper localization of MDC1 to sites of DNA damage.

**ID3 controls the interaction between γ-H2AX and MDC1.** The domains of MDC1 identified in this study as important for interactions with ID3 (Fig. 1) overlap with those previously identified as important for interactions with γ-H2AX[17]. Therefore, we explored whether the lack of ID3 or H2AX would affect interactions between MDC1 and γ-H2AX. We found that in ID3 knockdown cells, but not H2AX knockdown cells, the association between MDC1 and γ-H2AX in response to IR irradiation was abolished (Fig. 6a, b), indicating that the interaction between MDC1 and γ-H2AX occurs downstream of the ID3–MDC1 interaction.

To more closely examine the structural differences in MDC1 when interacting with either ID3 or ID3 and γ-H2AX together, we used molecular dynamic (MD) simulations to compare the interactions. We observed that the most stable conformation of the MDC1–tBRCT domain, as derived from normal MD simulations[28], is similar to that determined through X-ray diffraction data, and the salt bridge network including Arg1933, Glu2063, and Thr2067 involved in the binding of γ-H2AX is also in agreement (Supplementary Fig. 8a)[17, 29]. Interestingly, the distal C-terminal region of the tBRCT domain of MDC1 contains two positively charged lysine residues at positions 2071 (K2071) and 2075 (K2075) that are fully flexible, indicating that they may be relevant for binding to phospho-ID3. To explore this further, K2071 and K2075 were sequentially replaced with the neutral amino acid methionine in order to create both single-point and double-point mutants. The K2071M mutant bound to ID3-phosphorylated HLH (pHLH) equally as well as wild-type MDC1 (Fig. 6c). On the other hand, interactions between the K2076M mutant and ID3-pHLH were significantly decreased. Intriguingly, when both lysine residues were mutated, interactions were almost completely abolished. These results suggest a prominent role for K2075 in interactions with phospho-ID3, and perhaps a supportive role for K2071.

We then performed a docking simulation to more closely analyze the interactions between the tBRCT domain of MDC1 and the pHLH domain of ID3. Our simulation indicated the formation of a new hydrophobic core including Val67 of ID3 as a ligand and Phe1979, Ala2078, and Phe2079 of tBRCT as a receptor. The Cβ−methyl group of Thr2067 in tBRCT was located close to this cooperative hydrophobic core (Supplementary

Fig. 8b). Thus, Arg1933 in the tBRCT domain, which is known to be involved in binding to the tail of γ-H2AX[17], was much more flexible when in a complex with pHLH than it is in tBRCT alone. Consequently, it is likely that the weakness in the salt bridge of the tBRCT–pHLH complex may allow for better capture of the γ-H2AX pentapeptide. Of note, the tBRCT–γ-H2AX–pHLH trimer contains a substantial salt bridge network with aggregation of hydrophobic side chains, including Phe1979, Ala2078, and Phe2079 of tBRCT and Val67 of pHLH (Fig. 6d), indicating that the stability of the salt bridge increased when γ-H2AX docked with the tBRCT–pHLH complex. Together, these results suggest a molecular mechanism by which the interaction of MDC1–tBRCT with ID3-pHLH increases binding affinity between MDC1–tBRCT and γ-H2AX.

**ID3 promotes DSB repair through its interaction with MDC1.** To define a possible role for ID3 in DDR, we first investigated whether cells lacking ID3 would be more sensitive to DNA damage. Cell viability, as measured by the clonogenic survival assay, was measured for HeLa cells with and without depletion of ID3 through introduction of each of two different shRNAs. Under normal conditions, the ID3 knockdown cells were equivalent to wild-type cells. However, the cells depleted in ID3 had a significant decrease in survival in response to IR (Fig. 7a). Moreover, the survival fractions of colonies following IR were much lower in ID3 KO cells than WT cells (Supplementary Fig. 6d).

We next investigated whether ID3 affects DSBs repair by measuring γ-H2AX staining and comet tail moments. We found that knockdown of ID3 in HeLa cells had significantly more residual DSBs than control cells, as evidenced by the increase in signal intensity of γ-H2AX staining (Fig. 7b) and by the increase in comet tail moments (Fig. 7c). Notably, depletion of both ID3 and MDC1 did not further increase comet tail length from what was observed with ID3 depletion alone (Fig. 7d, e), indicating that ID3 and MDC1 act in the same pathway. We also confirmed that the impaired DSB repair was due to an insufficient amount of ID3, because ID3 knockdown cells could be rescued through the introduction of shRNA-resistant ID3 (Fig. 7f, g). These results suggest that ID3 promotes DSB repair.

Because we observed a role for ID3 in DSB repair, we predicted that a possible reason for the interaction between ID3 and MDC1 might also be to function in DNA repair. To test this, we stably transfected HeLa cells with ID3 shRNA in order to deplete endogenous ID3, reconstituted these cells with shRNA-resistant ID3-HLH or ID3-ΔHLH (Fig. 7h), and then measured the amount of DSB repair after IR exposure using the neutral comet assay. As shown in Fig. 7i, comet tails for cells depleted in ID3 or expressing ID3-ΔHLH were significantly longer than for those expressing ID3-HLH, suggesting that efficient DNA repair required the presence of the HLH domain. Further, we found that reconstitution of ID3-depleted cells with the S65E HLH mutant, but not the S65A HLH mutant, restored DSB repair

**Fig. 4** ID3 depletion reduces MDC1 foci formation. **a** HeLa and U2OS cells were transiently transfected with either control siRNA or ID3-specific siRNA. western blotting using antibodies to the indicated proteins shows the expression levels of each. **b, c** Control or ID3-depleted HeLa (**b**) and U2OS (**c**) cells were exposed to 2 Gy of IR and fixed at the indicated time points. Immunostaining experiments were performed using an anti-MDC1 antibody. Nuclei were stained with DAPI. Representative images (upper panel) and quantification (lower panel) of MDC1 foci in control and ID3-depleted cells (*n* = 3). Scale bars: 10 μm. **d, e** A stable knockdown of ID3 in HeLa cells co-expressing shRNA-resistant GFP-tagged wild-type (WT) ID3 and HA-MDC1 was generated. Levels of exogenous ID3 and MDC1 were confirmed by immunoblotting using the indicated antibodies (**d**). Images depict representative nuclei showing MDC1 foci at 1 h after IR treatment (**e**). Scale bars: 10 μm. The histogram in the lower panel is a quantification of the average number of cells containing MDC1 foci 1 h after exposure to IR (*n* = 3). **f, g** Control or ID3-depleted HeLa (**f**) and U2OS (**g**) cells were exposed to 2 Gy of IR. Images depict representative nuclei showing MDC1, γ-H2AX, NBS1, BRCA1, 53BP1, RNF8, and RNF168 foci at 1 h after IR treatment. The lower panel shows the number of IR-induced foci in control and ID3-depleted cells (*n* = 3). Scale bars: 10 μm. Uncropped blots of this Figure accompanied by the location of molecular weight markers are shown in Supplementary Fig. 14. Data are presented as means ± s.d. *P* values are based on Student's two-tailed *t*-test: **P* < 0.01

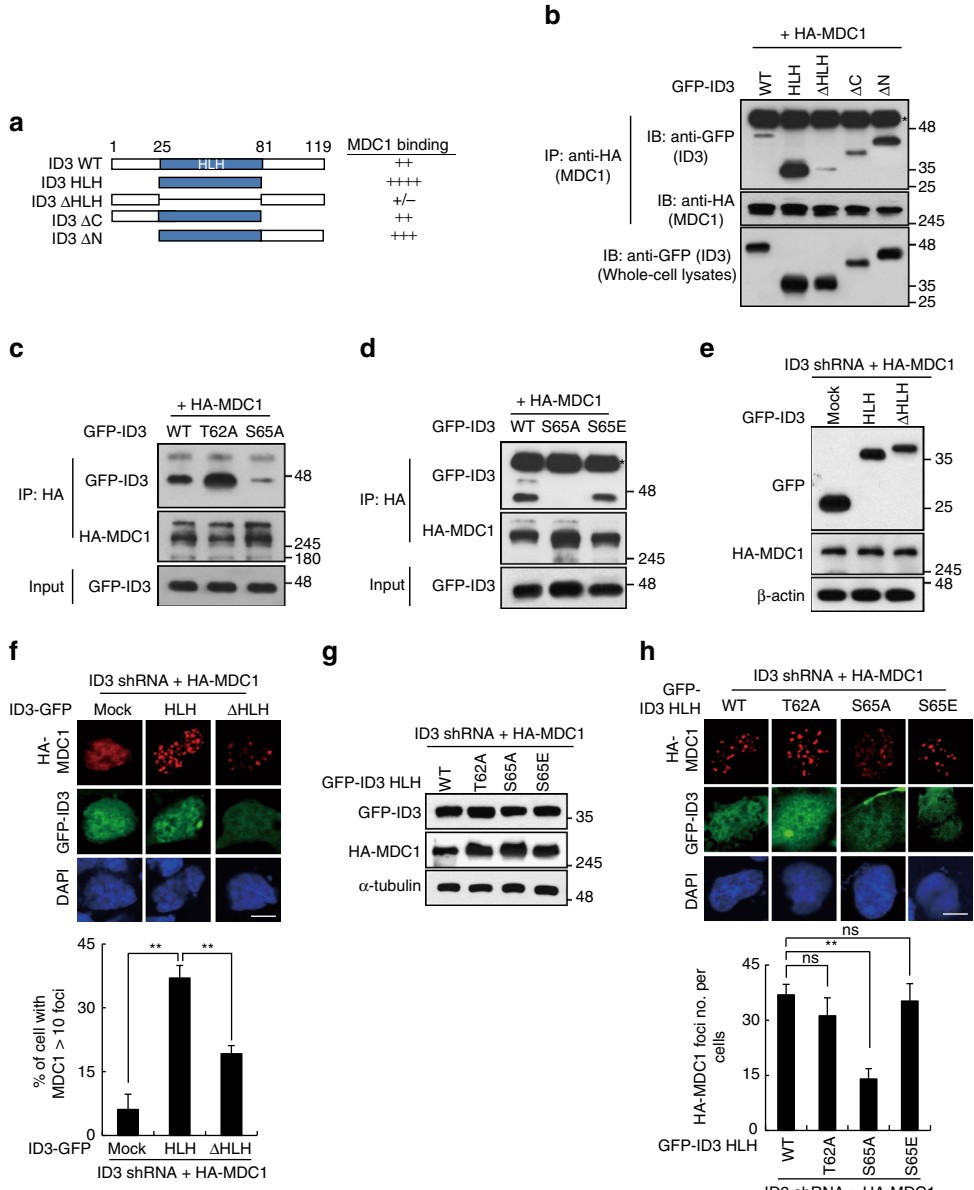

**Fig. 5** An interaction between ID3 and MDC1 is required for the recruitment of MDC1 at DSBs. **a** A schematic representation of wild-type ID3 (WT) and deletion constructs is shown. Domain boundaries, including the HLH region (25–81), are indicated using numbering for the human ID3 protein sequence. The relative affinity of MDC1 for the ID3 deletion mutants is indicated on the right side of the figure. **b** The HLH domain of ID3 is required for binding to MDC1. HEK293T cells were co-transfected with HA-tagged MDC1 and GFP-tagged deletion mutants of ID3, as indicated. After 48 h, cells were exposed to IR, and 3 h later, cell lysates were subjected to immunoprecipitation (IP) and immunoblotting (IB) using the indicated antibodies. * indicated heavy chain. **c** HEK293T cells were co-transfected with HA-MDC1 and one of three different versions of GFP-ID3: WT, T62A, or S65A. Cell lysates were subjected to IP using an anti-HA antibody and IB using antibodies indicated to the right of the image. **d** HEK293T cells were co-transfected with HA-MDC1 and one of three different versions of GFP-ID3: WT, S65A, or S65E. Cell lysates were subjected to IP using an anti-HA antibody and IB using antibodies indicated to the right of the image. **e**, **f** HeLa cells with a stable ID3 knockdown expressing HA-WT MDC1 were reconstituted with the indicated ID3 constructs or with a mock control. The exogenous ID3 and MDC1 were analyzed by IB using the indicated antibodies (**e**). Immunostaining for HA-MDC1 and GFP-ID3 was carried out 1 h after exposure to IR (**f**). Scale bars: 10 μm. The histogram in the lower panel quantifies the number of IR-induced foci (n = 3). **g**, **h** Similar to **e**, **f** but with the indicated point mutants in the HLH of ID3 expressed in stable ID3 knockdown HeLa cells (n = 3). Scale bars: 10 μm. Uncropped blots of this Figure accompanied by the location of molecular weight markers are shown in Supplementary Fig. 14. Data are presented as means ± s.d. P values are based on Student's two-tailed t-test: **P < 0.01. ns, not significant

(Fig. 7j, k), suggesting that phosphorylated S65 of ID3 is necessary. These findings established that the characteristics of ID3 that are important for interactions with MDC1 are also necessary to promote DSB repair.

Another intracellular role for MDC1 is to regulate intra-S-phase and G2/M cell-cycle checkpoints in response to DNA damage[30–32]. Thus, we looked for a role for ID3 in these cell-cycle checkpoints as well and found that an ID3 knockdown did not noticeably affect the G2/M checkpoints in response to IR (Supplementary Fig. 9). ID3 is involved in the transcriptional repression of p27, a key regulator of cell-cycle progression, and thus induced the G1/S transition[33, 34]. Indeed, HeLa cells

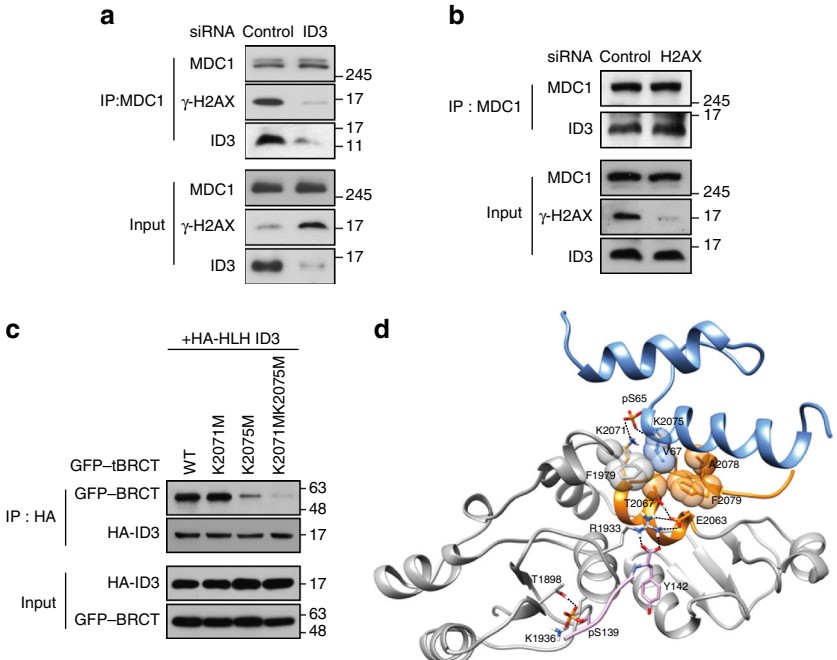

**Fig. 6** ID3 promotes binding of MDC1 to γ-H2AX. **a** Control and ID3-depleted HeLa cells were treated with and without exposure to IR. 3 h later, whole-cell lysates were subjected to immunoprecipitation (IP) using an anti-MDC1 antibody, and western blotting using anti-MDC1, anti-γ-H2AX and anti-ID3 antibodies as indicated to the right of the blot. **b** Control and H2AX-depleted HeLa cells were treated with and without exposure to IR. After 3 h, whole-cell lysates were subjected to immunoprecipitation using an anti-MDC1 antibody, and western blotting using anti-MDC1, anti-γ-H2AX, and anti-ID3 antibodies as indicated to the right of the blot. **c** GFP-tagged versions of each of the tBRCT point mutants were expressed in HEK293T cells together with the HLH domain of ID3. About 3 h after irradiation, cell lysates were subjected to immunoprecipitation and western blotting using antibodies as indicated to the right of the blot. **d** A model of the three-dimensional protein structures for the proposed trimer of MDC1–tBRCT, γ-H2AX, and ID3-pHLH is shown. The blue ribbon represents phosphorylated ID3-HLH, the gray ribbon represents the tBRCT domain of MDC1, the orange ribbon represents the region of tBRCT where ID3 binds, and the pink ribbon represents the γ-H2AX pentapeptide. Relevant amino acid residues are numbered, and notable hydrogen bonding is indicated by a black dashed line. Uncropped blots of this Figure accompanied by the location of molecular weight markers are shown in Supplementary Fig. 14

transfected with ID3 siRNA were less proliferative compared to control siRNA-transfected cells (Supplementary Fig. 10). This may be explained by the fact that IR-induced G2/M checkpoints were not affected in ID3-deficient cells.

**ID3 promotes DSB repair and genomic stability**. DSBs can be repaired through one of two pathways, HR or NHEJ. Because we had established a role for ID3 in promoting DNA repair, the next step was to look more closely at specific pathways using well-established reporter assays for both HR and NHEJ[35, 36]. We found that ID3 depletion led to decreased HR repair (Fig. 8a, b). NHEJ repair was also attenuated by knockdown of ID3 (Fig. 8c, d). Knockdown of ID3 did not cause an additional reduction of HR (Supplementary Fig. 11a) and NHEJ repair (Supplementary Fig. 11b) in MDC1-depleted cells, suggesting that ID3 regulates HR and NHEJ repair through MDC1.

Next, we measured the number of metaphase spreads containing chromosomal breaks that result from exposure to IR. As shown in Fig. 8e, cells in which ID3 had been depleted had an increased number of chromosomal breaks following IR treatment when compared to control cells. We then performed array-comparative genome hybridization (array CGH)[37], comparing the profiles of ID3-depleted and wild-type human fibroblast GM00637 and human embryonic lung fibroblasts MRC-5 cells. We found that chromosomal abnormalities were detected as chromosomal gains (dots over +0.5) and losses (dots below −0.5) that were widely distributed throughout the entire genome of ID3-depleted cells GM00637 (Fig. 8f and Supplementary Fig. 12)

and MRC-5 cells (Supplementary Fig. 13), indicating that the DSB repair deficiency observed when ID3 is depleted results in a failure to maintain chromosome integrity.

## Discussion

ID3 belongs to a family of proteins that consists of four members, ID1–ID4, which are ubiquitously expressed in many different tissues[38]. Most of the ID proteins contain a basic DNA recognition region that allows binding to either an E-box (CANNTG) or an N-box (CACNAG) in a promoter region; however, ID3 is lacking this region and instead, functions through dimerization with other transcriptional regulators[39]. Consequently, ID3 inhibits the activity of several transcription factors by directly binding and preventing their association with regulatory sequences in target genes[39–41]. ID3 plays an important role in controlling cell cycle, proliferation, differentiation, and apoptosis. ID3 has been also implicated as relevant to the pathology of vascular and metabolic disease[42–44]. These biological effects are mainly attributed to the aforementioned inhibition of transcription factors, specifically basic-HLH (bHLH) transcription factors. To our knowledge, the data presented here demonstrates for the first time that ID3 binds directly to proteins that are not transcription factors, and exerts its effects in a transcription-independent manner. We showed that ID3 directly binds to MDC1 and this interaction requires a tBRCT domain in MDC1 as well as an HLH domain in ID3. The BRCT domain of MDC1 is fairly conserved between human and murine proteins and it binds selectively to peptides containing a phosphorylated serine residue. Importantly,

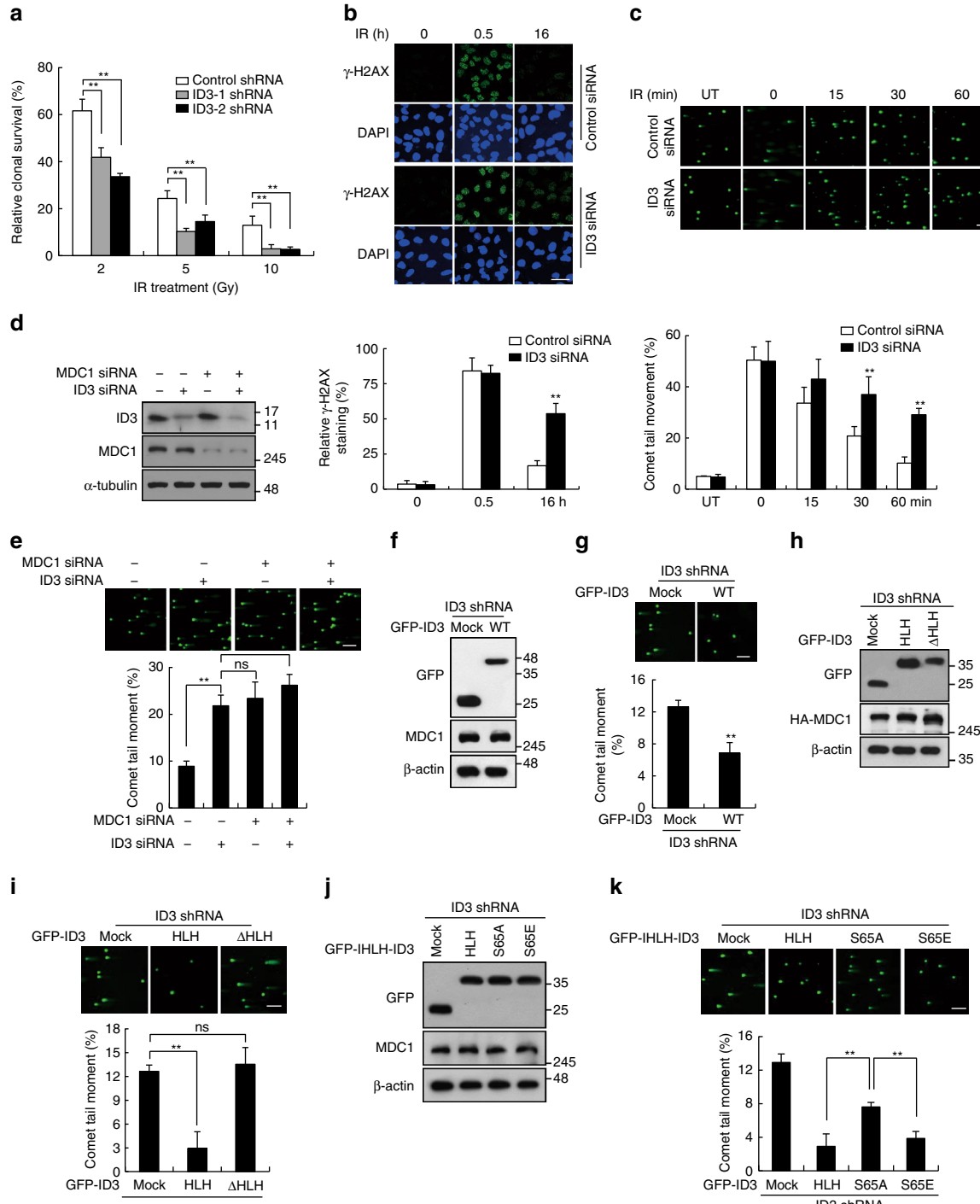

**Fig. 7** ID3-depleted cells are sensitive to IR and are defective in DSB repair. **a** HeLa cells with a stable ID3 knockdown were exposed to the indicated doses of IR and assessed for colony forming ability (n = 3). The cell viability of untreated cells is defined as 100%. **b**, **c** γ-H2AX foci (**b**) and comet moments (**c**) at indicated time points after exposure to IR in both control and ID3-depleted HeLa cells. Representative images (upper panel) and quantification (lower panel) of unrepaired DSBs are shown (n = 3). **d** HeLa cells were transfected with indicated siRNA combinations, and the expression levels of ID3 and MDC1 were assessed by western blotting using antibodies indicated. **e** HeLa cells were transfected with indicated siRNA combinations. 1 h after exposure to IR, the cells were assessed using the comet assay. Representative images (upper panel) and quantification (lower panel) of comet tail moments are shown (n = 3). **f** A western blot of GFP-tagged wild-type (WT) ID3 or GFP alone (mock construct) expressed in ID3 knockdown HeLa cells is shown. **g** Comet moments of complemented HeLa cells are shown as in (**e**). ID3-depleted cells were complemented with a GFP-WT ID3 construct or with GFP alone. **h** A western blot of the indicated GFP-ID3 constructs reconstituted into ID3 knockdown HeLa cells is shown. **i** Comet moments of complemented HeLa cells are shown as in (**e**). ID3-depleted cells were reconstituted with either HLH ID3 or an HLH deletion mutant (ΔHLH) of ID3 (n = 3). **j** A western blot demonstrating comparable expression of the indicated GFP-tagged point mutants in the ID3 HLH domain expressed in ID3 knockdown HeLa cells is shown. **k** Similar to **i** but with the indicated point mutants in the ID3 HLH domain expressed in ID3 knockdown HeLa cells (n = 3). Scale bars: 40 μm (**b**) and 200 μm (**c**, **e**, **g**, **i**, **k**). Uncropped blots of this Figure are shown in Supplementary Fig. 14. Data are presented as means ± s.d. P values are based on Student's two-tailed t-test: **P < 0.01. ns, not significant

a single phosphorylation of Ser65 in the HLH of ID3, is both necessary and sufficient for binding to MDC1. In addition, positively charged amino acid lysine at position 2075, which is fully flexible at the distal C-terminal region of tBRCT domain, plays an important role in association of MDC1–tBRCT with ID3–pHLH. ID1, ID2, and ID3 have a high degree of sequence similarity and have widely overlapping biological functions in many tissues[38]. However, ID3 is the only one of these three proteins that contains Ser65 motifs in the HLH domain, which we propose is the reason why ID1 and ID2 did not interact with MDC1. This suggests that ID3 may have a unique role in DNA repair that is distinct from the other ID proteins.

The direct and highly specific interaction between ID3 and MDC1 suggests that they are functionally linked. Consistent with this idea, we have shown that ID3 becomes phosphorylated at Ser65 in response to IR, and that the localization of ID3 to DSBs in response to DNA damage resembles that of MDC1. Moreover, our findings suggest that functional ID3 is required for MDC1 to localize appropriately after IR treatment. When we compared the effect of various mutant ID3 proteins, we showed that a phosphorylated Ser65 in the HLH domain was required for proper localization of MDC1 to damaged DNA sites. Furthermore, we demonstrated that when cellular levels of ID3 were low, there were fewer DSB repair events, HR and NHEJ activities were suppressed, and cells are more sensitive to IR. Notably, ID3-deficient cells transfected with either wild-type ID3 or the HLH domain of ID3, but not ΔHLH ID3 or S65A ID3, fully rescued both IR-induced MDC1 foci and repair of DSBs. Thus, we conclude that ID3 contributes to the ability of MDC1 to regulate DNA damage repair by directly interacting with MDC1.

Our data also highlight the necessary role that ID3 plays in allowing binding between MDC1 and γ-H2AX, early DNA damage sensor protein. MDC1 mainly acts as an adapter protein that recruits DDR proteins to sites of DNA damage[6]. After DNA damage, ATM is recruited to DSBs where it phosphorylates H2AX[1], and then γ-H2AX serves as a docking platform for MDC1, which binds to γ-H2AX via its tBRCT domain[17]. Abrogation of the MDC1-γ-H2AX interaction disrupts IR-induced

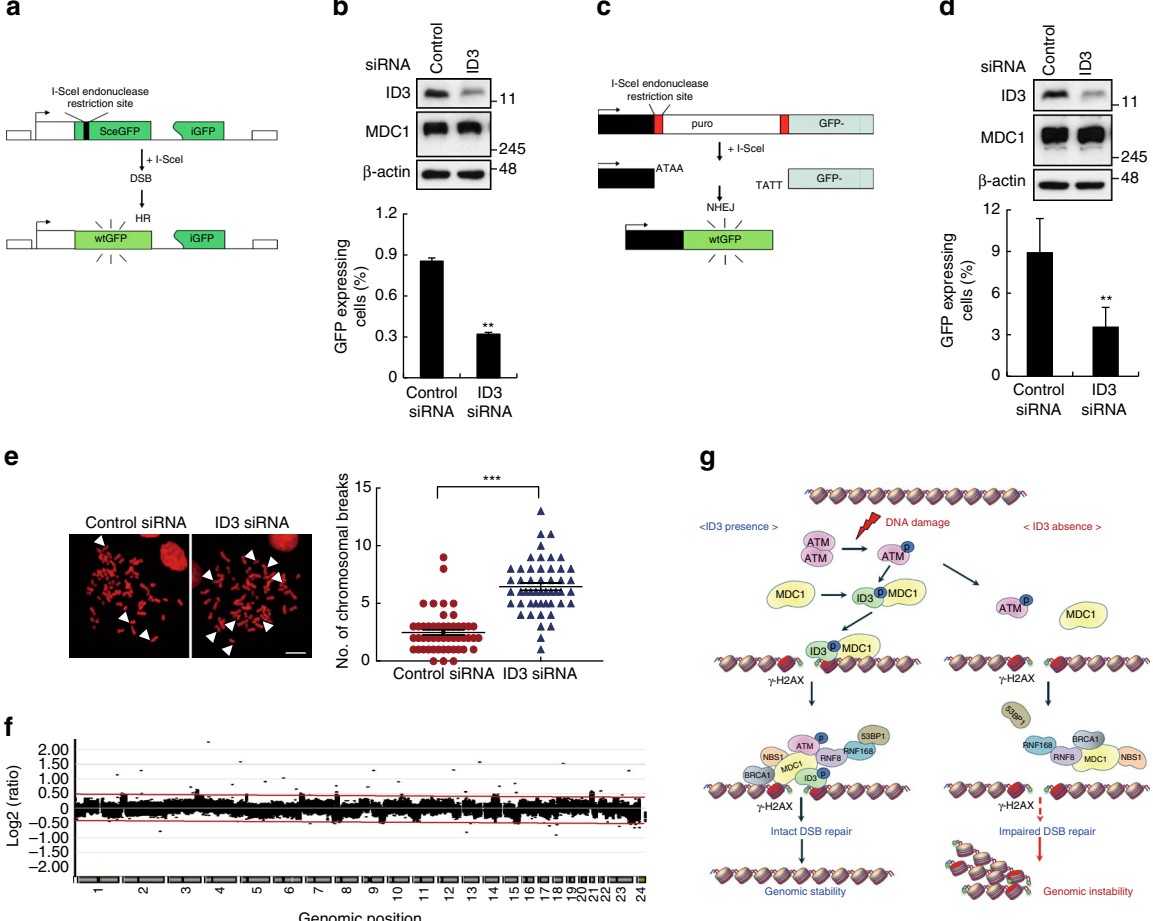

**Fig. 8** Efficiency of DSB repair in ID3-depleted cells. **a** A diagram of the fluorescence-based assay for measuring levels of DSB repair via homologous recombination (HR) is shown. **b** The efficiency of HR repair was measured by FACS analysis in DR-GFP-U2OS cells transfected with either control or ID3 siRNA (lower panel) ($n = 3$). The levels of endogenous MDC1 and ID3 were analyzed by western blotting (upper panel). **c** A diagram of the assay for measuring non-homologous end joining (NHEJ) repair using an EJ5-GFP reporter is shown. **d** The efficiency of NHEJ repair was measured by FACS analysis in HeLa cells that contained EJ5-GFP and had been transfected with either control or ID3 siRNA (lower panel) ($n = 3$). The levels of endogenous MDC1 and ID3 were analyzed by western blotting (upper panel). **e** Chromosome spreads from control or ID3-depleted HeLa cells exposed to IR are shown. Images (left panel) and a plot (right panel) of the average frequencies of chromosomal breaks ($n = 50$). Scale bars: 10 μm. **f** Array CGH profiles of clones derived from control or ID3-depleted GM00637 cells are shown. **g** A schematic representation of the role of ID3 in regulating the DDR functions of MDC1 is shown. The presence (left pathway) and absence (right pathway) of ID3 are compared. Uncropped blots of this Figure accompanied by the location of molecular weight markers are shown in Supplementary Fig. 14. Data are presented as means ± s.d. P values are based on Student's two-tailed t-test: **$P < 0.01$, ***$P < 0.001$

MDC1 foci formation and renders cells radiosensitive[45]. In this study, we observed the interaction between MDC1 and γ-H2AX was abolished by depletion of ID3. Our MD simulations suggest that interaction of MDC1–tBRCT with ID3-pHLH increases flexibility of Arg1933 of MDC1–tBRCT and therefore may give a better chance for catch the γ-H2AX. ID3 does not have obvious sequence homology with known DDR protein domains or relevant enzymatic active sites. Instead, ID3 may act as a scaffold for the MDC1-γ-H2AX complex by maintaining the interactions, stability, and DSB targeting of these proteins. Consequently, depletion of ID3 diminishes both the protein–protein interactions and the DSB-targeted activities of MDC1.

ID proteins are frequently overexpressed in many cancer cells, and disease severity and poor prognosis is associated with a high level of these proteins[41, 46, 47]. As a result, ID proteins are now considered important targets for potential anti-cancer drugs as a way to counteract tumor progression[41, 48]. However, a KO of ID3 resulted in the accumulation of DNA damage in colon cancer initiating cells[19], and conversely, ectopic expression of ID3 activated DNA repair processes in pancreatic β cells[18], thereby supporting a role for ID3 in maintaining genomic stability. Our studies revealed that ID3 depletion led to significant increases in DNA damage accumulation and chromosomal aberrations. The development of gross chromosomal abnormalities, which is a hallmark of cancer, in ID3-deficient non-transformed cells again supports the role of ID3 in maintaining chromosome stability. The association of ID3 with MDC1 and the effect on DSB repair that we have described in this study support our conclusion that ID3 functions to maintain genomic stability, very likely by modulating a DDR function of MDC1. Thus, although the inhibition of ID3 may suppress cancer cell proliferation and tumor progression, the impaired ability of MDC1 to repair DNA damage under these conditions might lead to a significant accumulation of chromosomal abnormalities and might subsequently promote tumorigenesis and cancer progression. This hypothesis is supported by a recent finding that ID3-KO mice develop T-cell lymphomas[49].

In summary, the interactions between ID3 and the DNA damage mediator protein MDC1 suggest that ID3 participates in the DNA damage signaling pathway to maintain genomic stability. The model depicted in Fig. 8g provides an integrated view of how MDC1 recruitment is linked to the upstream DNA damage signaling processes, and how ID3 may function at sites of DSBs. Our data establish a mechanistic basis for the ID3–MDC1 cooperation and suggest that, in response to IR, ID3 is phosphorylated by ATM and MDC1, specifically at Ser65 within the HLH domain, which then binds directly to MDC1–tBRCT domain, promoting both the interaction of MDC1 with γ-H2AX and the accumulation of MDC1 at DSB sites, leading to the recruitment of additional DDR factors at DSB sites. Thus, our results illuminate a novel interaction between ID3 and MDC1 that is crucial for DSB repair and genome stability.

## Methods

**Cell culture and treatment.** The human cervix carcinoma HeLa cells, human osteosarcoma U2OS cells and human embryonic kidney HEK293T cells were cultured in Dulbecco's modified Eagle's medium supplemented with 10% heat-inactivated fetal bovine serum (FBS: Lonza), 100 units per ml penicillin and 100 μg ml$^{-1}$ streptomycin (Invitrogen). The human fibroblast GM00637 and human embryonic lung fibroblasts MRC-5 cells were cultured in Earle's MEM containing 10% FBS, penicillin, and streptomycin. Cells were maintained in a humidified incubator with an atmosphere of 5% $CO_2$ at 37 °C. All cell lines were from the American Type Culture Collection (ATCC). To induce DNA DSB, exponentially growing cells were irradiated at 2 or 10 Gy from $^{137}$Cs source (Gammacell 3000 Elan irradiator, Best Theratronics) and allowed to recover at 37 °C incubator for various times. Inhibitors of ATM (Ku55933) and DNA-PKcs (Nu7426) were from TOCRIS bioscience.

**Antibodies.** Polyclonal MDC1 antibody (R2) was raised in rabbit against GST fusion protein containing BRCT domain of MDC1 (residues 1882–2089 aa). Antibody against S65 phosphorylated ID3 was generated to phospho-peptide of ID3 encompassing amino acids 61–72 (GTQLpSQVEILQR) using polyclonal antibody production service (Abfrontier). The antibodies used for immunoblotting analysis, immunoprecipitation assay and immunostaining assay are provided in Supplementary Table 1.

**ID3 siRNA and generation of stable ID3 knockdown cell lines.** For the knockdown of ID3, MDC1, or ATM expression, cells were transiently transfected with siRNA using lipofectamine RNAiMAX (Invitrogen) according to the manufacturer's instructions. The siRNA target sequences were as follows: ON-TARGETplus SMARTpool ID3 siRNA (ID3 CDS, NM_002167, GE Dharmacon), 5′-GCACUCAGCUUAGCCAGGU-3′, 5′-GAACGCAGUCUGGCCAUCG-3′, 5′-GGGAACUGGUACCCGGAGU-3′, 5′-GGAAGGUGACUUUCUGUAA-3′; MDC1 siRNA (MDC1 cDNA 58-76), 5′-UCCAGUGAAUCCAGGGUdTdT-3′; ATM siRNA (ATM cDNA 1266-1284), 5′-GAUACCAGAUCCUUGGAGAdTdT-3′; Negative control siRNA (Bioneer), 5′-CCUACGCCACCAAUUUCGUdTdT-3′. For generation of stable ID3-depleted cell lines, oligonucleotides encoding the target sequence for ID3-1- forward, 5′-GATCCCCACTGCTACTCC CGCCTGTTCAAGAGACAGGCGGGAGTAGCAGTGGTTTTTTG GAAA-3′; reverse, 5′-AGC TTTTCCAAAAAACCACTGCTACTCCCGCCTGTCTCTT-GAACAGGCGGGAGTAGCAGTGG G-3′ and ID3-2- forward, 5′-GATCCTCCTACAGCGCGTCATCGATTCAAGAGATCGATGA CGCGCTGTAGGATTTTTTGGAAA-3′; reverse, 5′-AGCTTTTCCAAAAAATCCTACAGCGCG TCATCGATCTCTTGAAT CGATGACGCGCTGTAGGAG-3′ were annealed and inserted into psilencer2.1-U6-neo vector (Ambion). Cells were transfected with pSilencer2.1-U6-neo control shRNA or pSilencer2.1-U6-neo ID3 shRNA using lipofectamine 2000 (Invitrogen) and cultured in selection medium containing 500 μg ml$^{-1}$ neomycin for 2–3 weeks. After selection, stably ID3 knockdown clones were confirmed by western blot analysis of ID3.

**Plasmid constructs and transfection.** The plasmids encoding wild-type MDC1 and various truncated MDC1 (Δ55–124, Δ200–420, Δ421–624, Δ625–1129, Δ1130–1661, Δ1893–2082) were obtained from Zhenkun Lou[50]. For GST pull-down assay, the BRCT fragment of MDC1 was produced by PCR using wild-type MDC1 (pcDNA HA-MDC1) as template and then subcloned in the EcoRI/XhoI site of pGEX4T-1 vector. To generate the full-length, N-terminal, C-terminal, HLH, ΔHLH constructs of ID3, each region of ID3 was amplified from human HeLa cDNA, and the PCR products were cloned into pEGFP-N3 or pcDNA HA (Fig. 2a; all amino acid positions were based on the sequence of accession NP_002158). To prepare the serial deletion constructs of tBRCT (1882–2089, 1882–2062, 1882–2042, 1882–2022, 1882–1992 (BRCT1), 1993–2089 (BRCT2), 2028–2089, 2063–2089), each fragment was PCR-amplified using pcDNA HA-MDC1 as template, and the PCR products were inserted into the EcoRI and XhoI sites of pEGFP-N3 vector. Point mutations of ID3 and tBRCT were generated by site-directed mutagenesis (Invitrogen). To construct the expression vector encoding shRNA-resistant GFP-tagged ID3, we performed mutagenesis using GEN-EART® Site-Directed Mutagenesis System (Invitrogen). The shRNA targeting sequence, 5′-CCACTGCTACTCCCGCCTG-3′, was replaced by 5′-CCACTGCTATTCCCGACTT-3′. All sequences were confirmed by automated DNA sequencing. Cells were transfected with the appropriate plasmids using lipofectAMINE 2000 (Invitrogen) according to the manufacturer's instructions.

**Immunoprecipitation assay and western blot analysis.** Cells were lysed in ice-cold RIPA buffer (50 mM Tris-HCl (pH 7.5), 150 mM NaCl, 1% Nonidet P-40, 0.5% sodium deoxycholate, 0.1% sodium dodecyl sulfate, 1 mM dithiothreitol, 1 mM phenylmethanesulfonyl fluoride, 10 μg ml$^{-1}$ leupeptin and 10 μg ml$^{-1}$ aprotinin). Equal amounts of cell or tissue extracts were separated by 6–12% SDS–PAGE followed by electrotransfer onto a polyvinylidene difluoride membrane (PALL life sciences). The membranes were blocked for 1 h with TBS-t (10 mM Tris-HCl (pH 7.4), 150 mM NaCl and 0.1% Tween-20) containing 5% non-fat milk and then incubated with indicated primary antibodies overnight at 4 °C. The blots were washed four times for 15 min with TBS-t and then incubated for 1 h with peroxidase-conjugated secondary antibodies (1:5000, Jackson ImmunoResearch Inc). The blots were washed four more times with TBS-t and developed using an enhanced chemiluminescence detection system (ECL; iNtRON Biotechnology, Korea). The amount of MDC1 protein was quantified using Scion Image software (Scion Corp.) For the immunoprecipitation assay, lysates were precleared with protein A-Sepharose beads (GE Healthcare) prior to adding the antibody. If DNase I or Ethidium Bromide was used, the lysates were either treated with 100 μg ml$^{-1}$ DNase I (Invitrogen) for 20 min at 37 °C or 50 μg ml$^{-1}$ ethidium bromide (Sigma) on ice for 30 min. Next, after removing the protein A-Sepharose by centrifugation, the supernatant was incubated at 4 °C overnight with appropriate antibodies (Supplementary Table 1). After the addition fresh protein A-Sepharose bead, the incubation was continued for an additional 1 h, and then beads were washed five times with RIPA buffer. Immunoprecipitated proteins were denatured in SDS

sample buffer, boiled for 5 min, and analyzed by western blotting using the appropriate antibodies (Supplementary Table 1).

**Immunofluorescence microscopy.** To visualize DNA damage foci, cells cultured on coverslips coated with poly-L-lysine (Sigma) were irradiated at 2 Gy and allowed to recover at 37 °C for adequate times. Cells were washed twice with PBS and fixed with 4% paraformaldehyde for 10 min and ice-cold 98% methanol for 5 min, followed by permeabilization with 0.3% Triton X-100 for 10 min at room temperature. After permeabilization, coverslips were washed three times with PBS and then were blocked with 5% BSA in PBS for 1 h. Cells were single or double immunostained with primary antibodies against the indicated proteins (Supplementary Table 1) overnight at 4 °C. Cells were washed with PBS and then stained with appropriate Alexa Fluor 488- (green, Molecular Probe), Alexa Fluor 594- (red, Molecular Probe) conjugated secondary antibodies (Supplementary Table 1). After washing, the coverslips were mounted onto slides using Vectashield mounting medium with 4,6 diamidino-2-phenylindole (Vector Laboratories, Burlingame, CA). Fluorescence images were taken using a confocal microscope (Zeiss LSM 510 Meta; Carl Zeiss) and analyzed with Zeiss microscope image software ZEN (Carl Zeiss). The foci number per cells was counted at least 100 cells. The error bars represent the standard error from three independent experiments.

**GST pull-down assay.** Bacterially expressed fusion proteins containing GST and the BRCT domains of MDC1 (residues 1893–2082) or GST alone immobilized onto Glutathion Sepharose 4B beads (GE Heathcare) and incubated with lysates prepared from HeLa cells or cells transiently transfected with GFP-ID3 expression vector for 3 h at 4 °C. Cells were lysed in TEN100 buffer (20 mM Tris-HCl (pH 7.4), 0.1 mM EDTA, 100 mM NaCl, 1 mM dithiothreitol, 1 mM phenylmethane-sulfonyl fluoride, 10 µg ml$^{-1}$ leupeptin, and 10 µg ml$^{-1}$ aprotinin). If λ-phosphatase treatment was required, TEN100 lysates were incubated with 400 units of λ-phosphatase (New England BioLabs) at 30 °C for 30 min. The GST beads were washed five times with NTEN buffer (0.5% NP-40. 20 mM Tris-HCl (pH 7.4), 1 mM EDTA, and 300 mM NaCl), and bound proteins were separated by SDS–PAGE and analyzed by western blotting using the appropriate antibodies.

**Clonogenic cell survival assay.** After treatment with irradiation, $5 \times 10^2$ cells were immediately seeded on 60 mm dish in triplicate and grown for 2–3 weeks at 37 °C to allow colonies to form. Colonies were stained with 2% methylene blue/50% ethanol and were counted. The fraction of surviving cells was calculated as the ratio for the plating efficiency of treated cells over untreated cells. Cell survival results are reported as the mean value ± s.d. for three independent experiments.

**Single-cell gel-electrophoresis (Comet assay).** DSB repair was visualized by neutral single-cell agarose-gel electrophoresis. Briefly, indicated cells were collected (~ 10$^5$ cells per pellet), mixed with low-melting agarose, and layered onto agarose-coated glass slides. The slides were maintained in the dark for all of the remaining steps. Slides were submerged in lysis solution (Cat.#4250-050-01,TREVIGEN® Instructions, Gaithersburg, MD, USA) for 1 h and incubated for 30 min in neutral electrophoresis solution (100 mM Tris, 300 mM Sodium Acetate at pH 9.0). After incubation slides were electrophoresed (~ 30 min at 1 V cm$^{-1}$ tank length), and then gently immerse slide in DNA Precipitation Solution (1.5 M NH$_4$Ac) for 30 min at room temperature. After air-dried, comet slide stained with SYBR green. Average comet tail moment was scored for 40–50 cells per slide using a computerized image analysis system (Komet 5.5; Andor Technology, South Windsor, CT, USA).

**HR assay.** To measure the HR repair, stable cell lines expressing DR-GFP reports were generated by transfection using lipofectamine 2000. Clones were selected in the medium containing 500 µg ml$^{-1}$ neomycin for 2 weeks and screened for significant induction of GFP-positive cells following infection with I-SceI expressing adenovirus. U2OS-DR-GFP cells were transfected with control or ID3 siRNA using lipofectamine RNAiMAX, and then infected with I-SceI-carrying adenovirus at an estimated MOI of 10. After 72 h, GFP-positive cells were measured by fluorescence-activated cell sorting (FACSCalibur, BD Biosciences). The acquired data was analyzed using CellQuest Pro software (BD Biosciences). The data are presented as the mean ± s.d. value in three independent experiments.

**NHEJ assay.** The NHEJ assay was measured in HeLa EJ5-GFP cells, using methods previous described[36]. EJ5-GFP contains a promoter that is separated from a GFP coding region by puromycin resistance gene, which is flanked by two I-SceI sites that are in the same orientation. When the I-SceI-induced DSBs is repaired by NHEJ in HeLa EJ5-GFP cells, the puro gene is removed, and the promoter is rejoined to the rest of the GFP expression cassette, leading GFP expression. HeLa EJ5-GFP cells were kindly provided by Dr. Kee at the University of South Florida. Similar to the above HR assay, HeLa EJ5-GFP cells were transfected with control or ID3 siRNA using lipofectamine RNAiMAX, and then infected with I-SceI-carrying adenovirus at an estimated MOI of 10. After 3 days, the percentage of GFP-positive cells which had repaired the DSBs generated by I-SceI was determined by flow

cytometry. For each analysis, 10,000 cells were processed and each experiment was repeated three times.

**Chromosomal aberration analysis.** For chromosomal aberration analysis, indicated transfected-U2OS cells were treated with 1 Gy of γ-ray for 24 h. To arrest cells in metaphase, 300 ng ml$^{-1}$ colcemid (Sigma) was added 4 h before cell collection. Colcemid depolymerizes microtubules and inhibits the formation of mitotic spindle. Cell were collected in 15 ml tubes, gently resuspended in 40% of culture media for 10 min at 37 °C, and then fixed in equivalent volume of a freshly prepared fixative solution (3:1 mixture of methanol/acetic acids, Carnoy's solution). After removal of supernatant, pellets were resuspended in fixative solution, dropped onto a cleaned glass slide and air-dried overnight. The slide was mounted in Vectashield with DAPI (Vector Laboratories). Metaphase images were captured using confocal microscope (Zeiss LSM 510 Meta; Carl Zeiss) and analyzed with Zeiss microscope image software ZEN (Carl Zeiss).

**CGH array and data analysis.** Human fibroblast GM00637 cells were stably transfected with control or ID3 shRNA, and genomic DNA was isolated using AccuPrep® Genomic DNA Extraction kit (Bioneer) according to the manufacturer's instructions. Array CGH analysis was performed using the NimbleGen Human CGH 12 × 135 K whole-genome tiling v3.1 Array (Agilent Technologies). Human genomic DNA (1 µg) from ID3-depleted cells and reference DNA samples from control cells were independently labeled with fluorescent dyes (Cy3/Cy5), co-hybridized at 65 °C for 24 h, and then subjected to the array. The hybridized array was scanned using NimbleGen's MS200 scanner (NimbleGen systems Inc.) with 2 µm resolution. Log2-ratio values of the probe signal intensities were calculated and plotted vs. genomic position using Roche NimbleGen's NimbleScan v2.5 software. Data are displayed and analyzed in Roche NimbleGen SignalMap software and CGH-explorer v2.55.

**G2/M checkpoint assay.** HeLa cells were transiently transfected with control or ID3 siRNA. After 48 h of transfection, cells were exposed to IR for 3 h and then placed in 100 ng ml$^{-1}$ nocodazole-containing media for 3 h, and the cells were collected and washed with PBS and then fixed with 1% formaldehyde for 10 min at 37 °C. The cells were chilled on ice for 1 min and then permeabilized with 90% methanol at −20 °C overnight. The fixed cells were washed with PBS and blocked with incubation buffer (0.5% BSA in PBS) for 10 min. The cells were stained with anti-phospho-histone H3(S10)-Alexa Fluor 647-conjugated antibody (Cell Signaling Technology, 9716) at a 1:10 dilution in incubation buffer for 1 h in darkness at room temperature, and the cells were then washed and resuspended in PBS containing 50 g ml$^{-1}$ propidium iodide. At least 10,000 cells were analyzed by fluorescence-activated cell sorting (FACSort, Becton Dickinson, San Jose, CA). The acquired data were analyzed using the CellQuest Pro software (Becton Dickinson).

**BrdU incorporation assay.** To determine the replication rate, control and ID3-depleted HeLa cells were seeded in a 48-well plate. After 24 h, 10 µM BrdU was added to the cells, and the culture was then incubated for 2 h at 37 °C. After fixation of the cells, immune complexes were formed using peroxidase-coupled BrdU-antibodies. Colored products were measured in a microplate reader at 405 nm with a reference wavelength at ~490 nm. The relative DNA synthesis was calculated as the absorbance of ID3-depleted cells from the absorbance of control cells. The data are presented as the mean ± s.d. from triplicate experiments.

**Generation of U2OS-ID3-knockout cells.** The ID3 CRISPR (Guide sequence insert: 5′-CGAGGCGGTGTGCTGCCTGTCGG)-3′ construct targeting exon 1 of the human ID3 gene were designed using the Cas9-Designer (http://www.rgenome.net/cas-designer)[51]. SpCas9 expression plasmids (500 ng) and sgRNA plasmids (500 ng) were transfected to $2 \times 10^5$ U2OS cells using 4D-nucleofector (Lonza) SE kit and program CM-104 in 20 µl Nucleovette Strips. Genomic DNA was isolated with the Nucleospin Tissue Kit (MACHEREY-NAGEL) 72 h post-transfection. Target sites were amplified with adapter primers using Phusion polymerase (New England Biolabs). The resulting deep sequencing libraries were subjected to paired-end sequencing with the MiSeq system (Illumina). After MiSeq, paired-end reads were joined by the Fastq-join. Fastq-joined files were analyzed using Cas-Analyzer (http://www.rgenome.net/cas-analyzer/) to obtain a mutation frequency in edited cells. For single-cell colony expansion, transfected cells were diluted and sorted with a density of 1 cell per well into 96-well-plates. After 7–10 days incubation, only single-cell colonies were screened visually under microscope. When colonies reached 70% confluence, the cells were transferred to 24-well-plates. Genomic DNA was isolated from each single colony. Sequences of ID3 target region were analyzed by a targeted deep sequencing in each single-cell colony and further confirmed by Sanger sequencing.

**Model system preparation.** The three dimensional (3D) structure of both ID3 protein (PDB code: 2LFH)[52] as ligand and MDC1/NFBD1-γ-H2AX complex (PDB code: 2AZM) as receptor are revealed by X-ray experiments. The MDC1–tBRCT domain of MDC1/NFBD1-γ-H2AX complex (residues 1891–2083) is defined to bind the γ-H2AX tail (residues 138–142) in the phospho-peptide recognition. For

the docking simulation, we need the isolated three systems; MDC1–tBRCT domain such as the receptor, γ-H2AX and ID3 such as the ligand. To prepare the ligand, the initial 25 residues of disordered N-terminal of ID3 protein are discarded. We used a total 44 amino acids (residues 25–68) to forms the HLH domain of ID3 protein and the residue number is different with that of the experimental source (NCBI Refseq: NM_002167.3). We assigned the new residue number, corresponded to the experimental data. The PDB residues 25–68 are agreement with NCBI sequences 40–83. To relax the neutralized systems in a cubic box of TIP3P water model, pre-equilibration is performed under NPT environment before the NVT process. Fully NVT equilibrium is performed until all properties of interest have stabilized. All simulation temperature is at 277 K (4 °C) like our experiments. The MD simulations are performed with amber99SB-nmr-ILDN force field with a phospho-Serine[53, 54].

**Docking and molecular dynamics calculation.** The docking protocol are used to explore the vast conformational space of ligands in a short time but the major drawback is the poor flexibility of protein, which is not permitted to adjust its conformation upon ligand binding. To predict more reliable protein-ligand complexes, MD simulation can deal with both ligand and protein in a flexible way, allowing for an induced fit of the receptor-binding site around the newly introduced ligand. To complement the computational methods, we perform their combined approach to provide reliable starting complex structures from docking calculations and to incorporate protein flexibility and analyze complex stability in MD simulation. In order to find the reliable complex in the 3D space, HEX protein docking server[55], and GROMACS software[56] patched with PLUMED 1.3[57] were used for docking and MD studies. From our biology experimental result, we reports that Lys2075 of the MDC1 BRCT domain binds phosphorylated serine (pSer65) of ID3 protein to enhance the binding affinity of the γ-H2AX pentapeptide and the newly special-binding site of the MDC1 BRCT domain is residues 2063–2089. To understand the complex conformation in the atomic level, the MDC1–ID3 complex is built to restraint the distance between Lys2075 of the receptor and pSer65 of the ligand and the angular search range about its own coordinate origin is set up in HEX docking server. Repeatedly docking calculation is performed by various angular docking parameters to search the possible docking model. The type of docking calculation is performed "shape-only". After getting the highest-scoring orientations by the docking calculation, the possible docking complex is solvated in the water molecules and is neutralized with Na$^+$ or Cl$^-$. Various docking models are performed using MD simulation to discriminate among conformations of different stability. During MD simulation, all complex models have a restraint condition between the specific binding residues as Lys2075 of the MDC1 and pSer65 of ID3. The restraint distance is about 4.0 Å and the restraint force is about 100 KJ mol$^{-1}$. The trimer of tBRCT-γ-H2AX-pHLH complex is calculated as same condition. Compared with the structural stability by the potential energy of system, we can predict one of possible docking systems and show the 3D structural model to explain the interaction with the receptor and the ligand.

**Statistical analysis.** Data in all experiments are represented as mean ± s.d. Two-tailed Student's t-test was conducted using GraphPad Prism software (GraphPad software, Inc.) and Excel (Microsoft). P values < 0.01 were considered statistically significant (**).

**Data availability.** The data that support the findings of this study are available from the corresponding authors upon reasonable request.

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

## Acknowledgements

We thank Zhenkun Lou for providing MDC1 deletion constructs and helpful comments on the manuscript. This work is supported by the National Research Foundation of Korea (NRF), funded by the Ministry of Science, ICT, and Future Planning (NRF-2015R1A5A2009070, NRF- 2017R1A2B2007557 and NRF-2017R1A2B2008064).

## Author contributions

J.-H.L., S.-J.P. and H.J.Y. designed the experiments; J.-H.L., S.-J.P., M.-J.K., S.M.J. and S.-Y.J. performed the experiments; G.H. and I.-Y.C. provided help for the experiments; C.K. and E.K. performed MD simulations; J.Y. and S.B. generated ID3 KO U2OS cells; J.-H.L., S.-J.P. and H.J.Y. analyzed data; J.-H.L., S.-J.P., E.K., S.B. and H.J.Y. wrote the manuscript.

## Additional information

**Competing interests:** The authors declare no competing financial interests.

