## [Peer Review File · Nature Communications]

Reviewers' comments:

Reviewer #1 (Remarks to the Author):

Major points

In this interesting study, Lee et. al. describe a potential role for Id3 in the cellular machinery that governs DNA damage sensing and repair. The authors determined that Id3 is phosphorylated upon DNA damage through an ATM dependent mechanism, that Id3 directly interacts with MDC1, and they have mapped the protein domains necessary for Id3/MDC1 interactions. Moreover, Id3 regulates MDC1 interactions with gammaH2AX. The authors' finding that a phosphorylated serine in Id3, not shared by Id1 or Id2, is required for DNA repair, is a rare demonstration of a unique role for Id3 outside of the immune system.

1. Because Id3 plays a significant role in replication, which is relevant to DNA damage acquisition, the replication rate of cells before and after Id3 depletion should be reported. Of relevance, Williams, et al. reported that Id protein destabilization in primary osteosarcomas led to p53-independent induction of the cyclin-dependent kinase inhibitor (CDKI) p21, and cell-cycle arrest.

2. The studies presented here were conducted primarily in HELA and U2OS cells, derived from cervical cancer and osteosarcoma respectively. The conclusion that Id3 is required for efficient DNA repair is difficult to reconcile however, with published reports that primary cervical cancer and osteosarcoma exhibit both high levels of Id3 and pronounced genomic instability. For this reason, additional studies in primary tumor cells or normal human cells expressing Id3 (e.g embryonic stem cells) that complement the array-comparative genome hybridization in normal fibroblasts (Fig. 8f) are recommended.

Williams, et al. USP1 deubiquitinates ID proteins to preserve a mesenchymal stem cell program in osteosarcoma. *Cell* 146, 918–930 (2011).

Minor point

Please note grammar/spelling errors on lines 117 and 191

Reviewer #2 (Remarks to the Author):

Remarks to the Author:

Review of "ID3, an inhibitor of DNA binding 3 protein regulates the MDC1-mediated DNA damage response in order to maintain genome stability", by Lee et al.

This manuscript concerns the identification of ID3, a factor more commonly known as a transcriptional regulator, as being required for the DNA damage response (DDR) pathway. The authors demonstrate that ID3 is likely phosphorylated by ATM, and that it interacts with the adapter protein MDC1, to facilitate double-strand break (DSB)-mediated signaling. Given that DNA DSB repair is required for genome stability, is tightly linked to cancer genesis and progression and is relevant for the current hot topic of gene editing this is decidedly a relevant manuscript for Nature Communications.

The authors provide a large body of solid data to support their claims. First, they identify ID3 in a Y2H screen to look for interactors with MDC1. They show that this interaction is enhanced by DNA damage. The authors then identify sites of phosphorylation on ID3 that they attribute to the DDR kinase, ATM. Phospho-specific Abs are generated to substantiate these claims. A bevy of knockdown experiments are then conducted to confirm the interaction with MDC1 and how the lack of this interaction affects the DDR. Additional experiments are then presented demonstrating the specific site of phosphorylation on ID3 (S65) and the requirement of phosphorylation at this site for its interaction with MDC1 and subsequent impact on the DDR. Finally, the authors utilize

knockdown experiments to show a frank deficit in DNA DSB repair and genome stability when ID3 expression is reduced.

In general, I'm quite supportive of this manuscript. The topic is certainly important and the authors have presented very strong evidence that they have identified a novel player in this pathway. I thought the data, overall, were quite compelling and believable; the authors are to be congratulated. I have made some comments below regarding improvements to the manuscript, but I want to stress that by and large these do not detract from my overall enthusiasm of this manuscript, which was quite high.

Comments:

line 104. " ... the association occurred in non-irradiated cells ...". The authors subsequently go to great lengths to demonstrate that this interaction is inducible and regulated by phosphorylation.. Do they have a hypothesis for how the constitutive binding is regulated (and why)?

Line 104. The enhancement is certainly much more evident in Fig. 1a than it is in 1b. Comments?

Line 104. The authors pretty much use 10 Gy for all their experiments. This is an extremely high dose that is not biologically relevant (unless you were standing next to Chernobyl). Are the effects as observable at lower doses? Some sort of dose response experiment would probably be good here.

Line 117. It is not clear to me why d625 and d1130 are such poor binders. Comments?

Line 144.. The response that the authors describe is slow.. Decades of radiation work has shown that the bulk (~80%) of DSBs are quickly repaired (< 30') by C-NHEJ. The ones that get repaired later are the dirty ended lesions and the ones that invoke HDR. Thus, the data suggest an effect principally on HDR and not C-NHEJ, which doesn't agree with a later experiment.

Line 165. Again, the authors are using 10 Gy for these experiments, which is a ton. A clinically relevant dose would be closer to 2 Gy. Is the effect still obvious?

Line 201. I liked this experiment.

Line 222. I also thought that these experiments were clean and compelling.

Line 236. My only big knock on this paper is that almost everything was done with inhibitors and/or with siRNA or shRNA knockdowns supplemented with complex rescue experiments. In this day of Cas/CRISPR the authors could presumably have generated some cleaner reagents that would have given them as good or better results, or?

Line 250. I agree. Nice data.

line 265. I'm not a big fan of molecular modeling and experiments done in silico. With that said, I thought that this part of the manuscript was relatively logical and led to a mechanistically more enlightening picture of what might be happening. Thus, I wouldn't have done this, but I'm glad the authors did.

Line 295. I do think that most of this section is hypothetical (although an elegant model). On the positive side, however, I think the hypothesis is reasonable and it is, in future work, eminently testable so I'd lobby to keep this data in.

Line 298. "lacking". Not to pick nits, but cells lacking ID3 are null cells. In all the experiments

presented here, the cells are, at best, reduced for ID3 expression.

Line 341. These experiments as presented look pretty good. I would have been happier, however, if the authors had used some positive &/or negative controls — i.e., cell lines that are known to be defective in C-NHEJ or HDR to confirm their observations and to be able to compare their observed defects to a known repair factor.

Line 355. How many metaphases were analyzed here? I could not find this information in the legend or M&Ms.

Point to Point Responses to the Review

Reviewers' comments:

Reviewer #1 (Remarks to the Author):

We thank for his/her comments/suggestions, which have helped to further improve the quality of our manuscript. We have revised our manuscript incorporating all the changes suggested by the reviewer.

Major points:

In this interesting study, Lee et. al. describe a potential role for Id3 in the cellular machinery that governs DNA damage sensing and repair. The authors determined that Id3 is phosphorylated upon DNA damage through an ATM dependent mechanism, that Id3 directly interacts with MDC1, and they have mapped the protein domains necessary for Id3/MDC1 interactions. Moreover, Id3 regulates MDC1 interactions with gammaH2AX. The authors' finding that a phosphorylated serine in Id3, not shared by Id1 or Id2, is required for DNA repair, is a rare demonstration of a unique role for Id3 outside of the immune system.

1. Because Id3 plays a significant role in replication, which is relevant to DNA damage acquisition, the replication rate of cells before and after Id3 depletion should be reported. Of relevance, Williams, et al. reported that Id protein destabilization in primary osteosarcomas led to p53-independent induction of the cyclin-dependent kinase inhibitor (CDKI) p21, and cell-cycle arrest.

Response:

As suggested by the reviewer, the number of cells in the S-phase of the cell cycle was measured in the control siRNA- and ID3 siRNA-transfected HeLa cells by BrdU incorporation and used as an indicator of proliferation. As expected, knockdown of ID3 decreased in the population of cells in the in S-phase. We have provided this result as an additional supplementary figure (Revised supplementary Figure 9).

Supplementary Figure 9 Effect of ID3 downregulation on S-phase entry

HeLa cells were transfected with either control siRNA or ID3 siRNA, and were treated with BrdU. DNA synthesis was assessed based on BrdU incorporation. Data are expressed as relative BrdU incorporation of the untreated control cells. Results are shown as mean \pm SD (n = 3), ** $P < 0.01$.

2. The studies presented here were conducted primarily in HELA and U2OS cells, derived from cervical cancer and osteosarcoma respectively. The conclusion that Id3 is required for efficient DNA repair is difficult to reconcile however, with published reports that primary cervical cancer and osteosarcoma exhibit both high levels of Id3 and pronounced genomic instability. For this reason, additional studies in primary tumor cells or normal human cells expressing Id3 (e.g embryonic stem cells) that complement the array-comparative genome hybridization in normal fibroblasts (Fig. 8f) are recommended. Williams, et al. USP1 deubiquitinates ID proteins to preserve a mesenchymal stem cell program in osteosarcoma. Cell 146, 918–930 (2011).

Response:

We thank the referee for this suggestion. To address this point, we have performed array CGH, comparing the profiles of control and ID3-depleted normal human MCR5 cells. We observed that, like immortal GM00636 cells, ID3–downregulating normal human (human embryonic lung fibroblasts MRC-5) cells also resulted in an increase in a high frequency of chromosomal abnormalities, including clonal amplifications and deletions in discrete regions (Revised Supplementary Figure 12).

Supplementary Figure 12 Genome-Wide DNA Copy Number Variation in ID3-Depleted MRC-5 Cells.

Using the copy number of control siRNA-transfected MRC-5 cells as a baseline, the relative changes in copy number for ID3-specific siRNA-transfected MRC-5 cells were measured. Chromosomal regions above or below the red dotted line indicate amplifications or deletions

of genomic positions, respectively.

Minor point:

Please note grammar/spelling errors on lines 117 and 191

Response:

We apologize for these grammar/spelling errors and we have corrected these mistakes in our revised version of the manuscript as highlighted in red.

Reviewer #2 (Remarks to the Author):

Review of "ID3, an inhibitor of DNA binding 3 protein regulates the MDC1-mediated DNA damage response in order to maintain genome stability", by Lee et al. This manuscript concerns the identification of ID3, a factor more commonly known as a transcriptional regulator, as being required for the DNA damage response (DDR) pathway. The authors demonstrate that ID3 is likely phosphorylated by ATM, and that it interacts with the adapter protein MDC1, to facilitate double-strand break (DSB)-mediated signaling. Given that DNA DSB repair is required for genome stability, is tightly linked to cancer genesis and progression and is relevant for the current hot topic of gene editing this is decidedly a relevant manuscript for Nature Communications. The authors provide a large body of solid data to support their claims. First, they identify ID3 in a Y2H screen to look for interactors with MDC1. They show that this interaction is enhanced by DNA damage. The authors then identify sites of phosphorylation on ID3 that they attribute to the DDR kinase, ATM. Phospho-specific Abs are generated to substantiate these claims. A bevy of knockdown experiments are then conducted to confirm the interaction with MDC1 and how the lack of this interaction affects the DDR. Additional experiments are then presented demonstrating the specific site of phosphorylation on ID3 (S65) and the requirement of phosphorylation at this site for its interaction with MDC1 and subsequent impact on the DDR. Finally, the authors utilize knockdown experiments to show a frank deficit in DNA DSB repair and genome stability when ID3 expression is reduced.

In general, I'm quite supportive of this manuscript. The topic is certainly important and the authors have presented very strong evidence that they have identified a novel player in this pathway. I thought the data, overall, were quite compelling and believable; the authors are to be congratulated. I have made some comments below regarding improvements to the manuscript, but I want to stress that by and large these do not detract from my overall enthusiasm of this manuscript, which was quite high.

We thank the reviewer for their time and positive evaluation of our study. We also appreciate the criticisms the reviewer notes. We have worked to provide further clarification of the justification of our results.

Comments:

line 104. "... the association occurred in non-irradiated cells ...". The authors subsequently go to great lengths to demonstrate that this interaction is inducible and regulated by phosphorylation. Do they have a hypothesis for how the constitutive binding is regulated (and why)?

Response:

We thank the reviewer for this comment. Phosphorylation of ID3 Ser65 is likely to be crucial for binding of ID3 to MDC1, because S65A mutation completely impaired interaction with MDC1 (Figure 5d). However, a small amount of phosphorylation in endogenous ID3 protein occurs under non-stressed condition (see Figure 2b, 2c and 2d), thus, ID3 may bind to MDC1

even before IR treatment.

Line 104. The enhancement is certainly much more evident in Fig. 1a than it is in 1b. Comments?

Response:

We agree with the reviewer's comment. We think that this phenomenon may be caused by Ab sensitivity, because this phenomenon does not occur in exogenous co-IP experiments using HA-tagged MDC1 and GFP-tagged ID3.

Line 104. The authors pretty much use 10 Gy for all their experiments. This is an extremely high dose that is not biologically relevant (unless you were standing next to Chernobyl). Are the effects as observable at lower doses? Some sort of dose response experiment would probably be good here.

Response:

We completely agree with the reviewer's comment. Following the reviewer's suggestion, we have performed additional experiments using 2 Gy irradiation. We have now changed several Figures (Figure 2c, 3a, 3b, 3d, 4b, 4c, 4f, 4g) to new data.

Line 117. It is not clear to me why d625 and d1130 are such poor binders. Comments?

Response:

We agree with the reviewer's comment. Every time we repeated this co-IP experiment, we got the same experimental result (poor binding in d625 and d1130). Therefore, although the MDC1 BRCT domain is critical for its binding with ID3, probably due to the unidentified three-dimensional structure of MDC1 protein, these regions (d625 and d1130) of MDC1 may have affected the interaction between ID3 and MDC1.

Line 144.. The response that the authors describe is slow.. Decades of radiation work has shown that the bulk (~80%) of DSBs are quickly repaired (< 30') by C-NHEJ. The ones that get repaired later are the dirty ended lesions and the ones that invoke HDR. Thus, the data suggest an effect principally on HDR and not C-NHEJ, which doesn't agree with a later experiment.

Response:

We thank the reviewer for this comment. In fact, IR-induced ID3 phosphorylation and its foci formation occurs within 30 min after 2 Gy irradiation. Thus, although ID3 is mainly involved in the regulation of HDR, it is likely that ID3 also might contribute to the regulation of C-NHEJ activity. We have now changed ID3 phosphorylation and ID3 foci data to better clarify the timing of ID3 phosphorylation and foci formation after irradiation (Figure 2c and 3d).

Line 165. Again, the authors are using 10 Gy for these experiments, which is a ton. A clinically relevant dose would be closer to 2 Gy. Is the effect still obvious?

Response:

We completely agree with the reviewer comment. As mentioned above, we have now changed these figures (Figure 3a, b, d).

Line 201. I liked this experiment.

Line 222. I also thought that these experiments were clean and compelling.

Response:

We thank the reviewer for this positive feedback.

Line 236. My only big knock on this paper is that almost everything was done with inhibitors and/or with siRNA or shRNA knockdowns supplemented with complex rescue experiments. In this day of Cas/CRISPR the authors could presumably have generated some cleaner reagents that would have given them as good or better results, or?

Response:

We agree with the reviewer that it is important to test whether ID3 plays an important role in controlling MDC1 function. To this end, we generated ID3 knockout (KO) human U2OS cells using the CRISPR/Cas9 genome-editing system. We designed five sgRNAs to target ID3 exon 1 (Supplementary Fig. 5a) and chose one ID3 sgRNA (ID3_sg1) due to high levels of transfection (Supplementary Fig. 5b). We obtained cells with homozygous deletion of exon 1 and Sanger sequencing confirmed that two types of frameshift indels were created in the targeting region of ID3 exon 1 in the KO cells, but not in the WT cells (Supplementary Fig. 5c). Consistent with the sequencing data, ID3 protein was undetectable in clones 1 and 2 (CRISPR-ID3-KO-1 and CRISPR-ID3-KO-2, respectively), as measured by Western blotting (Supplementary Fig. 5d).

To confirm the crucial role for ID3 in the recruitment of MDC1 to nuclear foci after DNA damage, we have now additional experiments using ID3 KO U2OS cells which show that:

- 1) IR-induced MDC1 foci are significantly reduced in ID3 KO cells than in control cells (Supplementary Fig. 6a).
- 2) After IR, the recruitment of MDC1 downstream factors such as NBS1, BRCA1, 53BP1, RNF8 and RNF168, but not upstream regulator γ -H2AX, to DNA damage sites was significantly reduced in ID3 KO cells (Supplementary Fig. 6b, 6c).
- 3) The ID3 KO cells displayed a significant decrease in survival in response to IR (Supplementary Fig. 6d).

Taken together, we conclude that ID3 plays an important role in controlling DDR function of MDC1.

a

	RGEN Target (5' to 3')	Position	Direction	GC Contents (% , w/o PAM)	Out-of-frame Score	Mismatches		
						0	1	2
ID3_sg1	CGAGGCGGTGTGCTGCCTGTCGG	401	+	70	61.7	1	0	0
ID3_sg2	AACGCAGTCTGGCCATCGCCCGG	424	+	65	61.8	1	0	0
ID3_sg3	TCAGCGGCTCCTCAGCTGCCGGG	461	-	70	67.9	1	0	0
ID3_sg4	ATGTCGTCCAGCAAGCTCAGCGG	477	-	55	64.4	1	0	0
ID3_sg5	CGCCTGCGGGAACTGGTACCCGG	516	+	70	76.6	1	0	0

b

	CRISPR Target (5' to 3')	Sample	total count	mutated count	mutated ratio (%)
ID3_sg1	CGAGGCGGTGTGCTGCCTGTCGG	Control	24332	4	0.02
		Cas9-treated	19629	12291	62.62
ID3_sg2	AACGCAGTCTGGCCATCGCCCGG	Control	24332	4	0.02
		Cas9-treated	17349	11023	63.54
ID3_sg3	TCAGCGGCTCCTCAGCTGCCGGG	Control	30943	43	0.14
		Cas9-treated	23159	14335	61.90
ID3_sg4	ATGTCGTCCAGCAAGCTCAGCGG	Control	30943	43	0.14
		Cas9-treated	21094	10331	48.98
ID3_sg5	CGCCTGCGGGAACTGGTACCCGG	Control	29520	8	0.03
		Cas9-treated	22329	3761	16.84

c

d

Supplementary Figure 5 Generation of ID3 KO cell lines using CRISPR-Cas9.

(a) A diagram of the strategy for knocking out ID3 using the CRISPR-Cas9 (upper panel) and the CRISPR-Cas9 target sequences for ID3 gene knock-out (lower panel). Target sequences are highlighted in gray. Targets were selected within exon 1 of the ID3 coding sequence (CDS) region for complete knock-out mutation. Targets which had no potential off-target bearing 1-2 mismatches compared to the on-target were chosen. Cas9-Designer (<http://www.rgenome.net/cas-designer/>) was used for CRISPR target selection.

(b) Target efficiency of five sgRNAs targeting ID3. Targeted-deep sequencings were performed with genomic DNA isolated from Cas9 and sgRNA-treated U2OS cells and wild type cells as a control for each sgRNA. Cas9-Analyzer (<http://www.rgenome.net/cas-analyzer/>) was used for analysis of deep sequencing data.

(c) Sanger sequencing results of each ID3 knock-out (KO) U2OS cell strain. CRISPR-ID3 KO-1 had 5 bp deletion and CRISPR-ID3 KO-2 had 1 bp insertion as a homogeneous mutation. Both mutations caused a codon frame-shift that led to premature stop codons within ID3 coding sequences.

(d) CRISPR/Cas9-mediated inactivation of ID3 gene in two different KO clones of U2OS cells. Lack of ID3 protein expression was documented by Western blotting analysis. MDC1 and α -tubulin was included as an internal control.

a

b

c

d

Supplementary Figure 6 ID3 Knockout U2OS Cells Displayed Impaired MDC1 foci Formation and Increased Sensitivity to IR

(a) ID3 WT U2OS (ID3-WT) and two different ID3 KO U2OS (CRISPR-ID3 KO-1 and CRISPR-ID3 KO-2) cells were exposed to 2 Gy of IR and fixed at the indicated time points. Immunofluorescent staining was performed using an anti-MDC1 antibody. Nuclei were stained with DAPI. Representative images (upper panel) and quantification (lower panel) of the number of MDC1 foci in WT and ID3 KO cells. Data are reported as mean \pm SD ($n=3$), ** $P < 0.01$

(b) Western blotting using antibodies to the indicated proteins shows the expression levels of each in ID3 WT and KO U2OS cells.

(c) ID3 WT and KO U2OS cells were exposed to 2 Gy of IR. Images depict representative nuclei showing MDC1, γ -H2AX, NBS1, BRCA1, 53BP1, RNF8 and RNF168 foci at 1 h after IR treatment. The lower panel shows the number of IR-induced foci in WT and ID3 KO cells. Results are shown as mean \pm SD ($n = 3$). ** $P < 0.01$.

(d) ID3 WT and KO U2OS cells were exposed to the indicated doses of IR and assessed for colony forming ability. The cell viability of untreated cells is defined as 100%. The experiment was carried out in triplicate. Bars represent standard error of the mean, ** $P < 0.01$.

Line 250. I agree. Nice data.

line 265. I'm not a big fan of molecular modeling and experiments done in silico. With that said, I thought that this part of the manuscript was relatively logical and led to a mechanistically more enlightening picture of what might be happening. Thus, I wouldn't have done this, but I'm glad the authors did.

Response:

We thank the reviewer for this comment. If reviewer allows us to include the molecular modeling data in our manuscript, we would like to keep these data.

Line 295. I do think that most of this section is hypothetical (although an elegant model). On the positive side, however, I think the hypothesis is reasonable and it is, in future work, eminently testable so I'd lobby to keep this data in.

Response:

We appreciate the reviewer's comment and agree with the reviewer's comment.

Line 298. "lacking". Not to pick nits, but cells lacking ID3 are null cells. In all the experiments presented here, the cells are, at best, reduced for ID3 expression.

Response:

We apologize for this mistake. We have now changed "lacking ID3" to "depleted ID3".

Line 341. These experiments as presented look pretty good. I would have been happier, however, if the authors had used some positive &/or negative controls — i.e., cell lines that are known to be defective in C-NHEJ or HDR to confirm their observations and to be able to compare their observed defects to a known repair factor.

Response:

We thank the reviewer for this comment. In the new version of the figures we have include these controls (see new Supplementary Figure 10a and 10b).

Line 355. How many metaphases were analyzed here? I could not find this information in the legend or M&Ms.

Response:

We apologize to the reviewer for not being clear regarding this point. We analyzed metaphase spreads in 50 cells. In the new version of the manuscript, we have included a Figure 8e legend describing how many cells were used for chromosomal breaks analysis.

REVIEWERS' COMMENTS:

Reviewer #1 (Remarks to the Author):

The authors have identified a new mechanism by which Id3 regulates the DDR. They find that an MDC1-Id3 interaction is crucial to MDC1 recruitment to damaged sites. Further, the authors determine the protein domains required for the interaction and identification of a new phosphorylation site in Id3 that regulates MDC1 binding to gamma H2AX at damaged DNA. The authors have done an excellent job responding to reviewers' concerns and it is my opinion that the manuscript is ready for publication.

Reviewer #2 (Remarks to Author):

Remarks to the Author:

Review of "ID3, an inhibitor of DNA binding 3 protein regulates the MDC1-mediated DNA damage response in order to maintain genome stability", by Lee et al.

This is a revised manuscript concerns the identification of ID3, a factor more commonly known as a transcriptional regulator, as being required for the DNA damage response (DDR) pathway. The authors demonstrate that ID3 is likely phosphorylated by ATM, and that it interacts with the adapter protein MDC1, to facilitate double-strand break (DSB)-mediated signaling. Given that DNA DSB repair is required for genome stability, is tightly linked to cancer genesis and progression and is relevant for the current hot topic of gene editing this is decidedly a relevant manuscript for Nature Communications.

In the original version the authors provide a large body of solid data to support their claims. First, they identified ID3 in a Y2H screen to look for interactors with MDC1. They showed that this interaction is enhanced by DNA damage. The authors then identified sites of phosphorylation on ID3 that they attributed to the DDR kinase, ATM. Phospho-specific Abs were generated to substantiate these claims. A bevy of knockdown experiments were then conducted to confirm the interaction with MDC1 and how the lack of this interaction affects the DDR. Additional experiments were then presented demonstrating the specific site of phosphorylation on ID3 (S65) and the requirement of phosphorylation at this site for its interaction with MDC1 and subsequent impact on the DDR. Finally, the authors utilized knockdown experiments to show a frank deficit in DNA DSB repair and genome stability when ID3 expression is reduced.

I was very supportive of the original manuscript. I thought the authors had a novel, interesting observation and that they had done a quite good job of mechanistically explaining how ID3 could work. One of my bigger objections was the sole use of knockdown experiments to validate their claims and I asked them to possibly generate some KO lines to validate these results. In this manuscript they have provided those data. Admittedly, the phenotype of the Kos is not as striking as one might have hoped for (the IR sensitivity is mild), but in general they support all of the knockdown experiments and add depth to an already deep manuscript. Thus, I recommend accepting this version. Kudos to the authors for doing some fine science.

Minor points. There are some minor typographical errors scattered throughout the text, but presumably these can be editorial removed during the publishing process.

Point to Point Responses to the Review

Reviewers' comments:

Reviewer #2 (Remarks to the Author):

This is a revised manuscript concerns the identification of ID3, a factor more commonly known as a transcriptional regulator, as being required for the DNA damage response (DDR) pathway. The authors demonstrate that ID3 is likely phosphorylated by ATM, and that it interacts with the adapter protein MDC1, to facilitate double-strand break (DSB)-mediated signaling. Given that DNA DSB repair is required for genome stability, is tightly linked to cancer genesis and progression and is relevant for the current hot topic of gene editing this is decidedly a relevant manuscript for Nature Communications.

In the original version the authors provide a large body of solid data to support their claims. First, they identified ID3 in a Y2H screen to look for interactors with MDC1. They showed that this interaction is enhanced by DNA damage. The authors then identified sites of phosphorylation on ID3 that they attributed to the DDR kinase, ATM. Phospho-specific Abs were generated to substantiate these claims. A bevy of knockdown experiments were then conducted to confirm the interaction with MDC1 and how the lack of this interaction affects the DDR. Additional experiments were then presented demonstrating the specific site of phosphorylation on ID3 (S65) and the requirement of phosphorylation at this site for its interaction with MDC1 and subsequent impact on the DDR. Finally, the authors utilized knockdown experiments to show a frank deficit in DNA DSB repair and genome stability when ID3 expression is reduced.

I was very supportive of the original manuscript. I thought the authors had a novel, interesting observation and that they had done a quite good job of mechanistically explaining how ID3 could work. One of my bigger objections was the sole use of knockdown experiments to validate their claims and I asked them to possibly generate some KO lines to validate these results. In this manuscript they have provided those data. Admittedly, the phenotype of the Kos is not as striking as one might have hoped for (the IR sensitivity is mild), but in general they support all of the knockdown experiments and add depth to an already deep manuscript. Thus, I recommend accepting this version. Kudos to the authors for doing some fine science.

We thank the reviewer for their time and positive evaluation of our study.

Minor points: There are some minor typographical errors scattered throughout the text, but presumably these can be editorial removed during the publishing process.

Response:

We apologize for these typographical errors.

We have corrected all of these typographical errors, and the current manuscript does not have these errors.